# ATOM-Bench: From Atoms to Conclusions in Objective Evaluation of Large Multimodal Models Reasoning

## Abstract

Chain-of-Thought (CoT) reasoning has significantly enhanced the ability of Large Multimodal Models (LMMs) to tackle complex image–text tasks, establishing itself as a cornerstone of multimodal learning. Despite significant progress, the impact of CoT on LMMs still lacks objective evaluation and in-depth research. Current CoT evaluation paradigms rely on powerful LLMs as judges of free-form text, but this introduces bias and hallucination from the evaluator itself. Moreover, it may penalize models for stylistic variations rather than genuine reasoning failures, thereby undermining the fairness and reliability of the assessment. To address this gap, we introduce ATOM-Bench, a CoT evaluation framework built on objective atomic questions. ATOM-Bench decomposes complex reasoning tasks into a series of atomic nodes, covering 570 high-resolution real-world images and 2,920 questions across 4 cognitive dimensions, and 12 domains, including architecture, text, transportation, culture, climate, and geology. Our benchmark introduces three novel quantitative metrics to objectively analyze reasoning faithfulness, consistency, and robustness. Extensive experiments with 22 LMMs validate the effectiveness of our framework. The results reveal that even the strongest models often exhibit a mismatch between surface-level correctness of final answers and their underlying evidence comprehension, while also exposing cognitive rigidity when faced with objective facts.We believe that ATOM-Bench, as a more objective and diagnostic tool, will advance LMMs toward more reliable and faithful reasoning.

## 1 Introduction

With the development of large language models, Chain-of-Thought (CoT) prompting was introduced to tackle complex reasoning tasks in purely textual settings (Wei et al., 2022; Wang et al., 2022). This technique significantly improved models' ability to perform multi-step reasoning and problem-solving by guiding them through a sequence of coherent logical steps. Building on the success of CoT in text domains, large multimodal models (LMMs) have extended this approach into visual domains, achieving notable advances in image understanding, video reasoning, and geolocation (Guo et al., 2025; Dou et al., 2024; Hong et al., 2025; Xiaomi et al., 2025; Lu et al., 2025; Team et al., 2025). By leveraging both visual and textual cues within their reasoning chains, LMMs demonstrate considerable potential as general-purpose multimodal reasoners.

The prevailing CoT evaluation paradigm, which relies on a powerful LLM (Jiang et al., 2025; Qiang et al., 2025; Jiang et al., 2025) as a "judge" to score free-form reasoning, faces two key challenges:

- **Judge Models are also Unreliable.** LLM-based scoring strategy introduces a "Black-box evaluating a Black-box" problem, where the objectivity of the assessment is compromised by the judge model's own biases and potential for hallucination.
- **Biased Evaluation.** LLM-based scoring often conflates writing with reasoning, penalizing models for omitted clues due to style rather than capability, thereby hindering reproducibility and obscuring what is truly measured.

To address these limitations, we introduce ATOM-Bench, a process-oriented benchmark that evaluates CoT reasoning in LMMs using objective atomic questions. The dataset consists of 570 high-

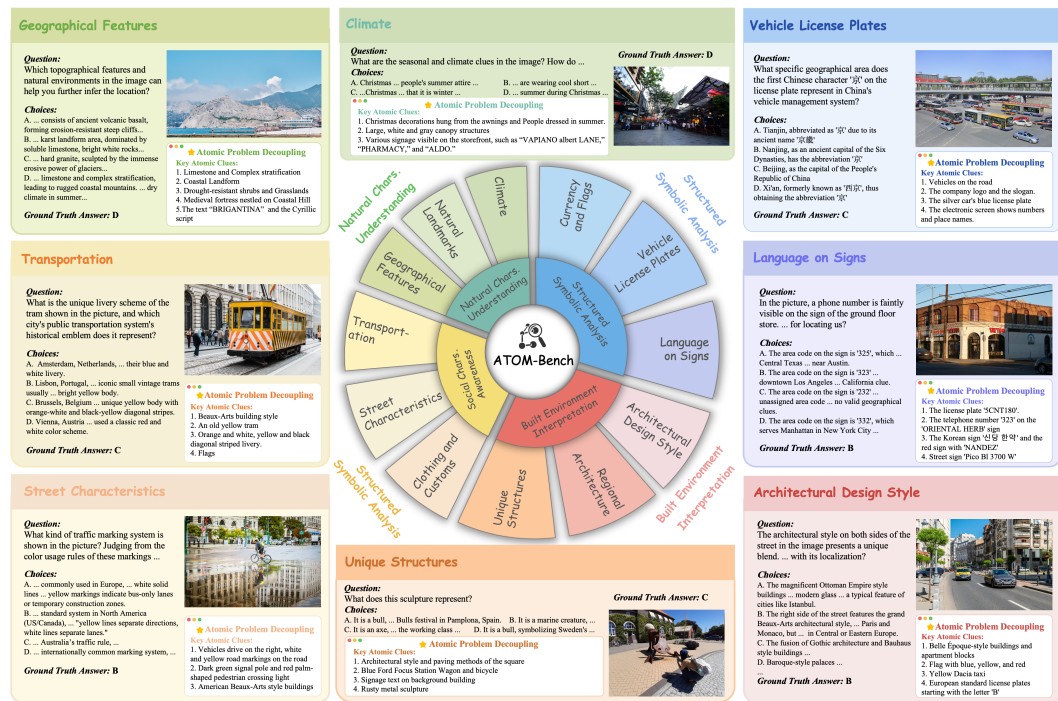

Figure 1: Typical samples from ATOM-Bench illustrate paired clue-level questions with images, alongside their decomposed atomic clues and corresponding GT annotations. ATOM-Bench evaluates models across diverse fields, ensuring a comprehensive evaluation of their capabilities.

resolution images, 1,696 annotated visual cues, and 2,920 multiple-choice questions across clue-level and result-level tasks. Each question has a unique ground-truth answer, enabling deterministic scoring without LLM judges. The questions are organized into 4 cognitive dimensions and 12 domains. By breaking complex tasks into atomic questions, ATOM-Bench enables fine-grained supervision, tracing how micro-decisions lead to final conclusions and shifting CoT evaluation to evidence-grounded, statistically robust diagnostics. We apply this framework to single-image geolocation for two reasons: 1) The task is fundamentally a complex reasoning problem, requiring LMMs to synthesize diverse visual cues from architecture to language and culture; 2) The logical steps in this reasoning process are inherently decomposable, allowing the construction of a ground-truth-based evaluation using objective atomic questions.

We decompose complex problems into atomic questions and employ 3 diagnostic metrics, RCS, HI, and RRS, to transform the vague CoT evaluation of LLM judges into clear and reproducible measures of reasoning behavior. By breaking down reasoning into verifiable subproblems, this framework overcomes the "black-box evaluating a black-box" pitfall of LLM-judge methods and avoids conflating writing fluency with reasoning ability or introducing narrative bias. RCS links conclusions to evidential support, HI reveals cases where correct answers rest on faulty evidence, and together they quantify the coupling between atomic-level accuracy and final predictions. RRS captures whether models can revise errors when confronted with ground-truth clues. Collectively, these metrics shift reasoning evaluation from subjective semantic matching to transparent, evidence-based diagnosis, providing a more reliable alternative to prior CoT assessment paradigms and objectively revealing whether models genuinely master the logical foundations of their predictions or merely rely on brittle heuristics.

Using ATOM-Bench, we evaluate 22 mainstream LMMs, finding that even state-of-the-art models misalign final accuracy with atomic reasoning and exhibit cognitive rigidity. Our contributions are:

- We propose a novel atomic-question-based CoT evaluation framework as an objective, reproducible alternative to the current mainstream paradigm of "LLMs judging CoT."

- We validate the framework on a carefully chosen, highly challenging task and construct the first high-resolution, process-oriented CoT benchmark.

- We evaluate state-of-the-art LMMs using our proposed metrics, revealing persistent gaps in reasoning faithfulness and cognitive flexibility, and offer insights. We hope this work will inspire future research toward more reliable LMMs.

## 2 RELATED WORK

### 2.1 MULTIMODAL LARGE LANGUAGE MODELS

Multimodal artificial intelligence has advanced rapidly (Radford et al., 2021; Li et al., 2022; OpenAI, 2023; 2024), extending language models (Touvron et al., 2023; Lin et al., 2023; Qwen Team, 2024; Liu et al., 2023b; Zhu et al., 2023; 2024; Dai et al., 2023; Xia et al., 2024c;d) to visual–language reasoning. Private models (OpenAI, 2024; 2023; Comanici et al., 2025; OpenAI, 2025b) exhibit strong visual understanding, but restrict broader innovation. Progress is driven by richer datasets (Chen et al., 2024a; Liu et al., 2024b; Wang et al., 2023; Ye et al., 2023), improved alignment (Dong et al., 2024; Li et al., 2024c; Liu et al., 2024a; Wang et al., 2024b), and efficient adaptation methods like LoRA (Hu et al., 2022). Open-source models (Liu et al., 2023a; Chen et al., 2024f; Bai et al., 2023; Yang et al., 2025; Zhu et al., 2025; Bai et al., 2025) further strengthen multimodal reasoning and are widely adopted. Recently, GPT-5 (OpenAI, 2025b), o1 (Jaech et al., 2024), and o3 (OpenAI, 2025c) highlight a shift toward enhanced reasoning, with works such as Wang et al. (2024c) improving robustness and Yao et al. (2024) identifying valid reasoning paths. Multi-step CoT approaches (Team, 2024; Du et al., 2025) echo o1-style iterative reflection and verification, yielding strong results in emerging LMMs (Guo et al., 2025; Hong et al., 2025; Xiaomi et al., 2025; Lu et al., 2025; Team et al., 2025).

Table 1: Dataset statistics.

| Statistic | Number | Percentage |
|---|---|---|
| **Total Images** | 570 | - |
| **Total questions** | 2920 | - |
| - Clue level questions | 1696 | 58.08% |
| - Location level questions | 1224 | 41.92% |
| - Multi-step reasoning | 3616 | 17.99% |
| Total Categories/Fields/Subfields | 2/4/12 | - |
| **Clue level questions fields:** | | |
| **- Built Environment Interpretation** | 450 | 26.53% |
| - - Architectural style | 230 | |
| - - Regional Architecture | 106 | |
| - - Unique Structures | 114 | |
| **- Social Characteristic Awareness** | 393 | 23.17% |
| - - Clothing and customs | 58 | |
| - - Street characteristics | 190 | |
| - - Transportation | 145 | |
| **- Structured Symbolic Analysis** | 617 | 36.38% |
| - - Currency and flags | 52 | |
| - - License plates | 102 | |
| - - Language on signs | 463 | |
| **- Natural Characteristic Understanding** | 236 | 13.92% |
| - - Geographical features | 186 | |
| - - Natural landmarks | 10 | |
| - - Climate | 40 | |
| Questions with Images | 2920 | 100% |
| Questions with answer label | 2920 | 100% |
| Average question length | 154.84 | - |
| Average option length | 90.94 | - |
| Average questions per image | 5.12 | - |

### 2.2 MLLM REASONING BENCHMARKS

While multimodal learning has advanced rapidly, evaluation benchmarks remain limited. Early work focused on perceptual abilities of LVLMs (Lu et al., 2022; Gurari et al., 2018), such as GQA (Hudson & Manning, 2019), but lacked depth for higher-order reasoning. More recent benchmarks broadened the scope: Golovneva et al. (2022); Prasad et al. (2023) evaluated reasoning chains in text, while others targeted scientific domains or final-answer tasks (Dou et al., 2024; He et al., 2024). General-purpose reasoning benchmarks, including MMStar Chen et al. (2024b), MMMU (Yue et al., 2024), and MME (Fu et al., 2023; Zhang et al., 2024c), extended evaluation to vision tasks, and later efforts such as LlamaV-o1 (Thawakar et al., 2025), MME-CoT (Jiang et al., 2025), MMIE (Xia et al., 2024b), and VER-Bench (Qiang et al., 2025) assessed multimodal reasoning chains and interleaved vision–language tasks. Yet, most adopt the "LLM-as-judge" paradigm (Li et al., 2024b; Zhang et al., 2023; Liu et al., 2023b;c; Yu et al., 2023; Chen et al., 2024c; Xia et al., 2024a; Jiang et al., 2024; Zhang et al., 2024a;b; Jiang et al., 2025), where stronger models semantically score free-form CoT. This "black-box evaluating a black-box" approach introduces bias, hallucination, and instability, undermining objectivity and reproducibility. To overcome this, we present ATOM-Bench, which decomposes complex reasoning into verifiable atomic problems, bypassing LLM judges and enabling transparent, objective, and reproducible evaluation.

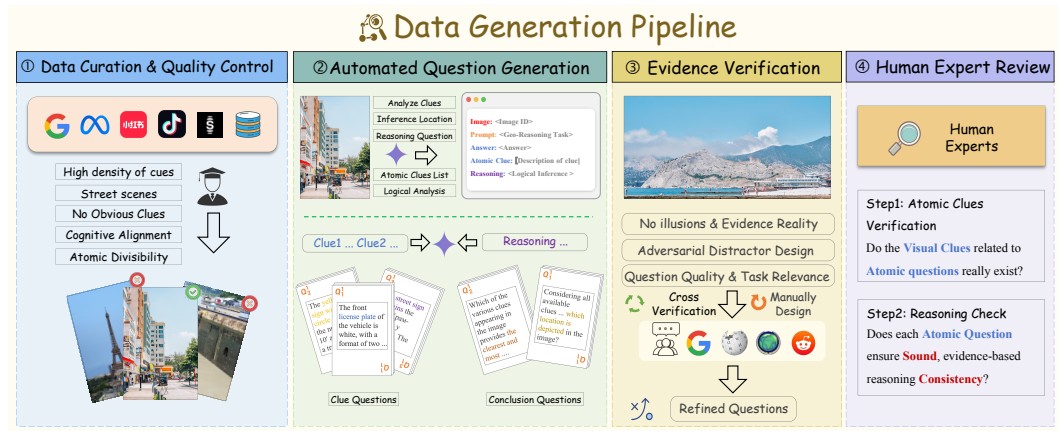

Figure 2: The construction of ATOM-Bench follows 4 steps: data collection; two-stage atomic question generation; rigorous human review and refinement; and final expert verification.

## 3 ATOM BENCH

### 3.1 OVERVIEW

In this section, we introduce ATOM-Bench, a benchmark for evaluating CoT reasoning in LMMs using atomic multiple-choice questions. Comprising 2,920 questions from 570 high-resolution images across 4 cognitive dimensions and 12 subtasks, ATOM-Bench enables evaluation of evidence understanding, reasoning faithfulness, and cognitive robustness. Unlike prior methods relying on LLM judges, it offers an objective framework for diagnosing LMMs. Dataset statistics are in Table 1.

### 3.2 ANNOTATION PIPELINE AND REVIEW

The images in ATOM-Bench are collected from public and online sources, primarily depicting real-world scenarios. As shown in Fig. 2, benchmark construction followed a systematic pipeline. The annotation team included six domain experts: 4 doctoral students and 2 senior research scientists.

**Step1: Data Curation and Quality Control.** Our data collection aimed to find images reflecting real-world complexity and rich in reasoning cues. We gathered 5,000 high-resolution images from public datasets and social media, prioritizing street scenes with dense details. Each image was manually reviewed for suitability to be deconstructed into atomic questions, ensuring diverse reasoning cues. After curation, 570 images were retained for annotation.

**Step 2: Automated Question Generation.** To create a challenging question set, we used a two-stage pipeline with Gemini-2.5-Pro. First, the model inferred the image's location and listed key visual clues, forming a preliminary evidence set. Second, these clues were used to generate question–option–answer triplets: (1) Clue-Level (CLQs), with three multiple-choice questions per clue to assess fine-grained understanding, and (2) Conclusion-Level (CoLQs), with 2–3 per image, asking for the location or the most decisive clue. Prompt templates are in Appendix E.

**Step 3: Answer and Visual Evidence Verification.** This stage involves a human-in-the-loop verification to refine automatically generated question–answer–evidence triplets, ensuring objectivity, accuracy, and challenge. Despite Gemini-2.5-Pro's strong generation capabilities, its black-box nature introduces errors, hallucinations, and post-hoc fallacies, necessitating human validation. Expert annotators reviewed 15 candidate questions per image based on three principles: (1) evidence grounding:cited clues must be visibly present and externally verified; (2) adversarial distractor design:options crafted from real but irrelevant text, confusable alternatives, or hallucinations as "bait"; (3) quality and relevance:coherent, task-aligned questions are retained, with low-quality ones rewritten. This ensures all dataset questions are human-vetted with high value. Details are in Appendix C.

**Step 4: Final Expert Cross-Validation.** In the final annotation stage, expert annotators cross-validated all QA pairs by (1) verifying the presence of visual clues and eliminating hallucinations, and (2) ensuring the reasoning linking clues to answers was logically sound and aligned with real-world knowledge. These steps ensure each atomic question is a valid part of the reasoning chain.

Table 2: **Accuracy results on ATOM-Bench.** Best scores are highlighted in **bold**, with orange for private models, blue for open-source models, and green for open-source thinking models.

| Model | LLM | $CLQ_{Acc}$ | $CoLQ_{Acc}$ | RCS | HI | RRS | Avg |
|---|---|---|---|---|---|---|---|
| *Private Models* | | | | | | | |
| Gemini-2.5-Pro (Comanici et al., 2025) | - | **74.16** | **84.42** | **65.96** | **31.24** | **67.06** | **72.07** |
| GPT-5 (OpenAI, 2025b) | - | 65.87 | 78.89 | 55.98 | 41.37 | 59.51 | 63.78 |
| GPT-4.1 (OpenAI, 2025a) | - | 63.65 | 77.68 | 55.63 | 42.28 | 59.00 | 62.74 |
| *Open-Source Models* | | | | | | | |
| Qwen2.5-VL-72B (Bai et al., 2025) | Qwen2.5-72B | **61.51** | **70.91** | **49.82** | **46.98** | 24.53 | **51.96** |
| InternVL3-78B Zhu et al. (2025) | Qwen2.5-72B | 60.44 | 65.17 | 46.99 | 48.44 | 20.04 | 48.84 |
| Qwen2.5-VL-32B (Bai et al., 2025) | Qwen2.5-32B | 57.77 | 67.67 | 46.10 | 50.10 | 18.51 | 47.99 |
| InternVL3.5-38B (Wang et al., 2025) | Qwen3-32B | 56.13 | 58.86 | 41.67 | 53.56 | 17.59 | 44.14 |
| InternVL2.5-78B (Chen et al., 2024d) | Qwen2.5-72B | 52.43 | 58.64 | 39.36 | 55.78 | 15.55 | 42.04 |
| InternVL3-8B Zhu et al. (2025) | Qwen2.5-7B | 49.78 | 59.14 | 37.05 | 58.78 | 19.43 | 41.32 |
| InternVL2.5-38B (Chen et al., 2024d) | Qwen2.5-32B | 48.98 | 53.53 | 34.22 | 60.29 | 8.62 | 37.01 |
| Llama-3.2-11B-Vision (Dubey et al., 2024) | Llama-3.1-8B | 47.99 | 54.06 | 34.57 | 60.84 | 6.88 | 36.53 |
| Qwen2.5-VL-7B (Bai et al., 2025) | Qwen2.5-7B | 43.69 | 57.72 | 29.08 | 67.20 | 18.61 | 36.38 |
| LLaVA-OneVision-72B (Li et al., 2024a) | Qwen2-72B | 45.73 | 47.23 | 30.67 | 62.23 | 13.00 | 34.88 |
| InternVL3.5-8B (Wang et al., 2025) | Qwen3-8B | 43.25 | 46.59 | 28.37 | 65.22 | 11.98 | 32.99 |
| MiniCPM-o2.6-8B (Team, 2025) | Qwen2.5-7B | 40.88 | 46.65 | 28.73 | 66.25 | 12.49 | 32.50 |
| InternVL2.5-8B (Chen et al., 2024d) | InternLM-7B | 43.66 | 48.83 | 26.58 | 68.02 | 8.52 | 31.91 |
| InternVL2-8B (Chen et al., 2024e) | InternLM-7B | 36.51 | 36.42 | 21.63 | 72.27 | 5.56 | 25.57 |
| LLaVA-OneVision-7B (Li et al., 2024a) | Qwen2-7B | 34.94 | 40.24 | 19.49 | 74.71 | 2.91 | 24.57 |
| Qwen2-VL-7B (Wang et al., 2024a) | Qwen2-7B | 8.25 | 35.57 | 11.81 | 84.28 | 6.99 | 15.67 |
| *Open-Source Thinking Models* | | | | | | | |
| GLM-4.1V-9B-Thinking (Hong et al., 2025) | GLM-4-9B-0414 | 60.58 | 65.69 | **47.52** | **48.76** | **18.10** | **48.63** |
| MiMo-VL-7B-SFT-2508 (Xiaomi et al., 2025) | MiMo-7B | 57.15 | **65.77** | 44.50 | 51.54 | 17.08 | 46.59 |
| Ovis2.5-9B (Lu et al., 2025) | Qwen3-8B | 56.91 | 63.85 | 42.73 | 52.09 | 16.37 | 45.55 |

## 3.3 DATA COMPOSITION AND CATEGORIZATION

To diagnose LMMs' reasoning in complex real-world scenarios, ATOM-Bench introduces a multi-dimensional taxonomy. Unlike prior domain-based benchmarks, it deconstructs geospatial reasoning into fundamental cognitive abilities. We posit that reliable reasoning requires mastery of multiple, largely orthogonal dimensions. All atomic questions in ATOM-Bench are mapped into 4 categories:

**Built Environment Interpretation.** Captures reasoning over human-made structures, including architectural styles, regional architecture, and unique structures, reflecting cultural, historical, and geographic constraints essential for localization.

**Social Characteristic Awareness.** Encompasses human activities and customs visible in the environment, including clothing and customs, street characteristics, and transportation. Such cues reveal socio-cultural context often decisive for geolocation.

**Structured Symbolic Analysis.** Evaluates the ability to decode formal symbol systems tied to explicit rules, such as currency and flags, license plates, and language on signs. These structured cues link directly to geopolitical and institutional knowledge.

**Natural Characteristic Understanding.** Involves reasoning about the natural world governed by physical and environmental principles. Fine-grained cues include geographic features, natural landmarks, and climate, which provide strong geographic signals.

The final dataset contains 2,920 questions and 1,696 visual cues from 570 diverse images, spanning street scenes, sports events, natural landscapes, and urban architecture. Covering 12 clue types, it offers high information diversity, enabling comprehensive evaluation across cognitive dimensions.

## 4 AUTOMATED EVALUATION METRIC

Evaluating LMMs requires moving beyond final-answer accuracy to diagnosing their CoT. Existing CoT evaluations, often judged by stronger LLMs, introduce bias and conflate reasoning with writing ability. ATOM-Bench addresses this by decomposing complex reasoning into structured atomic questions, enabling objective evaluation of both reasoning faithfulness and cognitive robustness.

### 4.1 CoT REASONING FAITHFULNESS

This component quantifies the logical consistency between a model's final conclusion and its grasp of the atomic evidence supporting it. It addresses the key question: To what extent is the final

answer grounded in correct evidence comprehension? A faithful model should only reach the correct conclusion through accurate understanding of each necessary logical step. As detailed in Section 3.2, each instance $i$ includes an image $I_i$, one conclusion-level question $Qr_i$, and $N_i$ clue-level questions $Qc_i = \left\{ qc_i^{(1)}, \ldots, qc_i^{(N_i)} \right\}$. For a LMM, let $\mathcal{M}(q, I_i)$ denote its answer to question $q$. The result-level answer correctness $CoLQ_i$ and the clue-level answer accuracy $CLQ_i$ are defined as:

$$CoLQ_i = \mathbb{I}(\mathcal{M}(Qr_i, I_i) \text{ is correct}), \quad CLQ_i = \frac{1}{N_i} \sum_{j=1}^{N_i} \mathbb{I}\Big(\mathcal{M}(qc_i^{(j)}, I_i) \text{ is correct}\Big) \tag{1}$$

We introduce the **Reasoning Consistency Score (RCS)** to quantify the likelihood that a model reaches the correct conclusion for the right reasons. For dataset $\mathcal{D}$, RCS is the expected joint event where the conclusion is correct and clue accuracy exceeds a threshold $\tau$ ($\tau=0.75$):

$$\text{RCS} = \mathbb{E}_{i \in \mathcal{D}} \left[ CoLQ_i \cdot \mathbb{I}(CLQ_i > \tau) \right] \tag{2}$$

To capture post-hoc fallacies and pattern-based guessing, we define the **Hallucination Index (HI)**. HI measures the conditional probability that, in the model's correct conclusion cases, its conclusion is correct while the clue-level accuracy falls below threshold $\tau$. Formally, let $\mathcal{D}$ denote the set of instances where the conclusion is correct ($CoLQ_i = 1$), and $\mathcal{B}$ is the set where clue accuracy does not exceed $\tau$ ($CLQ_i \leq \tau$) ($\tau = 0.75$):

$$\text{HI} = \frac{|\mathcal{D} \cap \mathcal{B}|}{|\mathcal{D}|} \tag{3}$$

High HI value provides a clear indication that the model's correct conclusions are largely decoupled from their evidentiary basis, revealing a significant lack of faithfulness in its reasoning process.

## 4.2 CoT Cognitive Robustness

This component evaluates the rigidity of a model's reasoning and its ability to correct errors when confronted with objective facts. We operationalize it through a "golden evidence cross-examination" experiment, asking: How resistant is a model's reasoning to revision under indisputable evidence?

The evaluation proceeds in two stages. First, the model answers the conclusion-level question $Qr_i$ for an image $I_i$, getting its first output $O_{first}$. Second, it is given the ground-truth clue set $GT_i$ and asked again, producing final output $O_{final}$. To capture robustness, we define the **Robust Reasoning Score (RRS)**, which assigns higher weight to corrections on harder questions. The dataset $\mathcal{D}$ is divided into $\mathcal{D}_{hard}$ ($n = 157$), $\mathcal{D}_{medium}$ ($n = 423$), and $\mathcal{D}_{easy}$ ($n = 644$), with respective weights of 3, 2, and 1. The difficulty level for each instance was determined based on a consensus among multiple annotators, where the final classification reflects the intersection of their judgments.

We define a behavioral value function $\beta$ to score the four possible outcomes of the experiment. Let $C_{first}$ and $C_{final}$ denote the correctness of the first answers $O_{first}$ and final answers $O_{final}$. $\beta$ assigns $+1$ when the model remains correct (steadfast correctness) or corrects an error (evidence-based correction), and $-1$ when it persists in error (cognitive rigidity) or shifts from correct to incorrect (logical confusion). The **RRS** for a model $\mathcal{M}$ on dataset $\mathcal{D}$ is then computed as the difficulty-weighted sum of $\beta$, normalized by the maximum achievable score.

$$\text{RRS} = \frac{\sum_{i \in \mathcal{D}} \omega(i) \cdot \beta(C_{first,i}, C_{final,i})}{\sum_{i \in \mathcal{D}} \omega(i)} \times 100\% \tag{4}$$

The **RRS** provides a holistic measure of cognitive robustness by capturing both a model's ability to correct errors and its performance across varying levels of task difficulty.

## 4.3 Accuracy Evaluation Strategy

To mitigate random guessing in the multiple-choice format, we adopt a **Penalty-Adjusted Scoring Mechanism**, inspired by standardized tests such as the SAT. Each correct answer yields 1 point, while an incorrect one incurs a penalty of $-\frac{1}{n-1}$, with $n$ denoting the number of choices; unanswered questions score 0. This ensures that random guessing has an expected score of zero, offering a fairer measure of model ability. The final standardized score $CoLQ_{Acc}$ and $CLQ_{Acc}$:

$$\text{CoLQ}_{\text{Acc}} = \frac{\sum_{i \in \mathcal{D}} \text{Score}(\mathcal{M}, Qr_i)}{|\mathcal{D}|}, \quad \text{CLQ}_{\text{Acc}} = \frac{\sum_{i \in \mathcal{D}} \text{Score}(\mathcal{M}, Qc_i)}{|\mathcal{D}|} \tag{5}$$

Table 3: **Evaluation Results of Clue Class Accuracy for Different Models in ATOM-Bench .** Best performance is marked in **bold** with `orange` for private models, `blue` for open-source models, and `green` for open-source thinking models.

| Model | Arch. Styles | Reg. Arch. | Unique Struct. | Cloth. & Cust. | Street Chars. | Trans-port. | Curr. & Flags | Lang-uage | Lic. Plates | Cli-mate | Geo-graphy | Nat. Land. |
|---|---|---|---|---|---|---|---|---|---|---|---|---|
| | *Built Environ.* | | | *Social Chars.* | | | *Struct. Symbol* | | | *Nat. Chars.* | | |
| *Private Models* | | | | | | | | | | | | |
| Gemini-2.5-Pro (Comanici et al., 2025) | **77.88** | **76.34** | **77.24** | 81.75 | **74.52** | **79.56** | **72.85** | **68.91** | **69.84** | **83.96** | **72.54** | **86.67** |
| GPT-5 (OpenAI, 2025b) | 73.43 | 74.06 | 64.44 | **81.90** | 64.74 | 67.66 | 59.87 | 59.18 | 53.34 | 70.83 | 70.52 | **86.67** |
| GPT-4.1 (OpenAI, 2025a) | 70.60 | 71.54 | 56.26 | 75.14 | 69.47 | 67.66 | 59.87 | 55.44 | 54.57 | 67.50 | 66.89 | **86.67** |
| *Open-Source Models* | | | | | | | | | | | | |
| Qwen2.5-VL-72B (Bai et al., 2025) | **70.02** | **63.92** | 51.73 | 77.16 | 60.53 | **64.25** | 52.34 | 55.84 | **59.39** | 64.17 | 66.22 | 73.33 |
| InternVL3-78B Zhu et al. (2025) | 69.99 | 49.21 | **56.55** | **81.90** | **61.32** | 55.99 | 52.18 | 55.55 | 52.93 | **64.17** | **70.38** | 73.33 |
| Qwen2.5-VL-32B (Bai et al., 2025) | 63.17 | 57.63 | 56.48 | 72.70 | 52.24 | 54.30 | **62.28** | **56.01** | 45.25 | 57.71 | 64.11 | **86.67** |
| InternVL3.5-38B (Wang et al., 2025) | 65.06 | 46.78 | 54.06 | 68.10 | 57.11 | 62.58 | 60.19 | 48.42 | 46.35 | 57.71 | 63.35 | **86.67** |
| InternVL2.5-78B (Chen et al., 2024d) | 63.86 | 57.86 | 44.82 | 70.26 | 56.40 | 49.74 | 47.05 | 45.27 | 43.94 | 63.75 | 54.17 | 46.67 |
| InternVL3-8B Zhu et al. (2025) | 56.98 | 42.92 | 36.71 | 72.84 | 53.67 | 48.58 | 47.05 | 41.26 | 47.90 | 57.71 | 63.35 | 60.00 |
| InternVL2.5-38B (Chen et al., 2024d) | 56.40 | 45.44 | 42.41 | 70.40 | 48.71 | 55.95 | 44.49 | 40.66 | 42.72 | 60.83 | 57.04 | 46.67 |
| Llama-3.2-11B-Vision (Dubey et al., 2024) | 52.99 | 47.72 | 45.99 | 75.00 | 39.65 | 46.83 | 52.02 | 45.94 | 37.52 | 47.71 | 51.88 | **86.67** |
| LLaVA-OneVision-72B (Li et al., 2024a) | 56.81 | 38.84 | 32.79 | 70.40 | 47.94 | 50.66 | 34.07 | 40.17 | 29.95 | 57.50 | 53.23 | 46.67 |
| InternVL3.5-8B (Wang et al., 2025) | 58.08 | 40.49 | 36.78 | 47.56 | 42.37 | 48.58 | 41.92 | 39.54 | 34.88 | 44.37 | 47.32 | 6.67 |
| Qwen2.5-VL-7B (Bai et al., 2025) | 49.91 | 42.69 | 41.05 | 56.75 | 36.08 | 44.21 | 47.05 | 42.46 | 46.59 | 34.17 | 43.90 | 46.67 |
| InternVL2.5-8B (Chen et al., 2024d) | 55.75 | 31.60 | 36.64 | 56.75 | 43.82 | 45.11 | 46.89 | 34.95 | 37.49 | 47.71 | 56.94 | 60.00 |
| MiniCPM-o2.6-8B (Team, 2025) | 45.83 | 41.59 | 38.71 | 54.31 | 40.35 | 41.43 | 42.08 | 36.10 | 41.17 | 37.71 | 44.09 | 33.33 |
| InternVL2-8B (Chen et al., 2024e) | 51.65 | 31.53 | 34.37 | 58.91 | 43.86 | 35.02 | 26.70 | 28.00 | 16.82 | 37.50 | 42.56 | 46.67 |
| LLaVA-OneVision-7B (Li et al., 2024a) | 42.99 | 30.42 | 28.45 | 52.30 | 37.53 | 33.33 | 36.79 | 27.94 | 27.03 | 31.04 | 45.43 | 60.00 |
| Qwen2-VL-7B (Wang et al., 2024a) | 8.93 | 7.94 | 7.40 | 20.11 | 4.81 | 0.37 | 13.88 | 7.95 | 0.89 | 20.83 | 13.26 | 33.33 |
| *Open-Source Thinking Models* | | | | | | | | | | | | |
| MiMo-VL-7B-SFT-2508 (Xiaomi et al., 2025) | 63.17 | 55.42 | 46.86 | 72.99 | 58.42 | 56.14 | 54.74 | 53.53 | 42.47 | **64.38** | 66.85 | 73.33 |
| Ovis2.5-9B (Lu et al., 2025) | **67.14** | 48.74 | 54.11 | **75.14** | 59.25 | **58.69** | 62.44 | 50.34 | 40.27 | 64.17 | 61.96 | **86.67** |
| GLM-4.1V-9B-Thinking (Hong et al., 2025) | 64.99 | **65.17** | **59.84** | 68.25 | **61.89** | 48.30 | **65.00** | **56.50** | **52.80** | 60.83 | **71.24** | 73.33 |

# 5 EXPERIMENTS

In this section, we evaluate various models on ATOM-Bench, addressing three questions: (1) Are conclusions grounded in evidence or driven by shortcuts? (2) Do models exhibit cognitive rigidity when presented with evidence? (3) Does process-level evaluation offer deeper insights than outcome-based scoring? Section 5.1 details the setup, and Section 5.2 presents results.

## 5.1 EXPERIMENT SETUP

**Evaluation Models.** To evaluate ATOM-Bench, we tested a wide range of models, including private models such as GPT-5 (OpenAI, 2025b), GPT-4.1 (OpenAI, 2025a) and Gemini-2.5-Pro (Comanici et al., 2025). We also assessed leading open-source models, including InternVL3.5 (8B, 38B) (Wang et al., 2025), Qwen2.5-VL (7B, 32B,72B) (Bai et al., 2025), Qwen2-VL (7B) (Wang et al., 2024a), InternVL3 (8B,38B) (Zhu et al., 2025), InternVL2.5 (8B,38B,78B) (Chen et al., 2024d), InternVL2 (8B) (Chen et al., 2024e) and MiniCPM-o-2.6 (8B) (Team, 2025), and LLaVA-OneVision (7B,72B) Li et al. (2024a). In addition, we included recent "thinking" models such as MiMo-VL-7B-SFT-2508 (Xiaomi et al., 2025), Ovis2.5-9B (Lu et al., 2025) and GLM-4.1V-9B-Thinking (Hong et al., 2025). Open-source models were run on 8 NVIDIA A100 40G GPUs via the MS-Swift framework, while private models were accessed through APIs.

## 5.2 FINE-GRAINED ANALYSIS AND FINDINGS

In this section, we present the comprehensive evaluation on ATOM-Bench . The detailed performance of LMMs is shown in Table 2 and Table 3. Our analysis and key findings are as follows:

**Faithful Reasoning Remains Elusive.** As shown in Table 2 and Fig. 3(c-d), all model families achieve higher accuracy on conclusion questions (CoLQ) than on clue questions (CLQ). For example, Gemini-2.5-Pro reaches 84.42% CoLQ but only 74.16% CLQ, and Qwen2.5-VL-72B records 70.91% and 61.51%. This gap suggests models are better at producing final answers than analyzing evidence. We quantify it with the evidence–conclusion spread ($ECS = CoLQ − CLQ$), where smaller values imply stronger evidential grounding. As shown in Fig.5, "thinking" models achieve relatively high CLQ despite smaller scale, suggesting process supervision benefits evidence com-

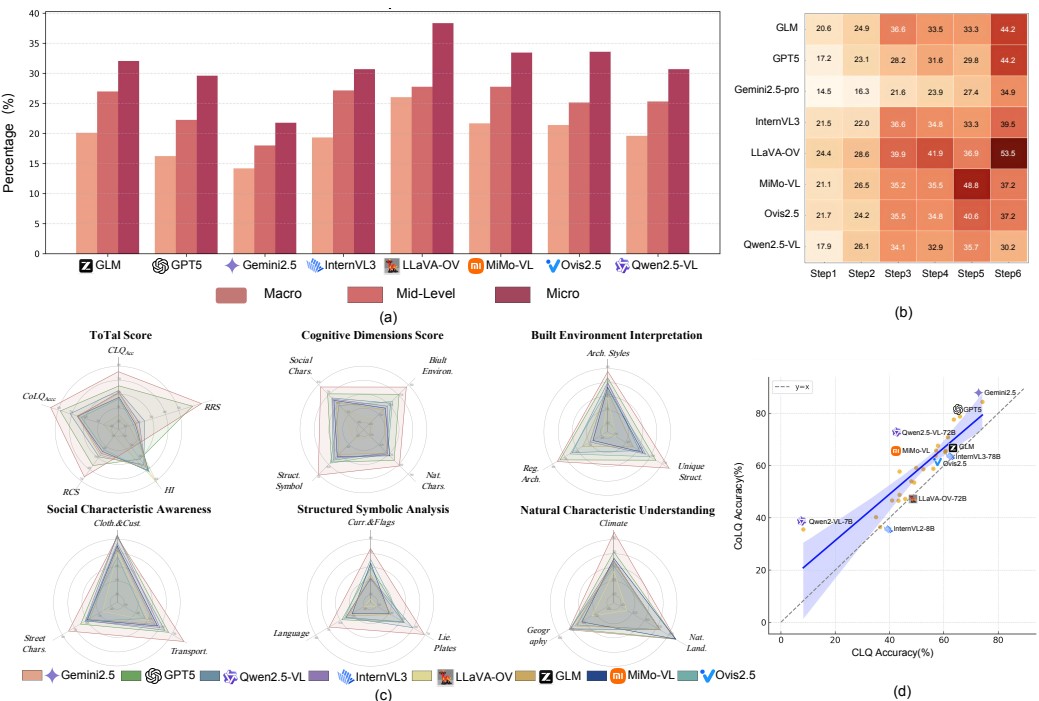

Figure 3: (a) Error rates grouped by scope level. (b) Matrix illustrating error frequencies across reasoning steps. (c) Model scores across 5 metrics, 4 cognitive domains, and 12 atomic clues. (d) Models tend to score higher on conclusions than evidence, indicating reliance on shortcut reasoning.

prehension more than scaling. Correlation analysis ($\rho$=0.91) confirms CLQ and CoLQ are generally aligned, while their negative correlation with ECS ($\rho$=−0.27) underscores that evidence understanding, not final accuracy, reduces the gap. Detailed results are in AppendixA.1.

**Models Fall into the Trap of Post Hoc Fallacy.** To strengthen our evaluation, we assess logical pitfalls through the Reasoning Consistency Score (RCS) and Hallucination Index (HI). As shown in Fig. 6 and Fig. 7, models with high CoLQ but low RCS often rely on unsupported conclusions, and larger evidence–conclusion gaps (ECS) correlate with higher hallucination rates. The joint distribution of RCS and HI exposes systematic **post-hoc** bias: models frequently treat correct answers as evidence of valid reasoning, even when the underlying causal chain is flawed. For example, GPT-5 achieves 78.89% CoLQ but only 55.98% RCS with 41.37% HI, while Qwen2.5-VL-72B shows a similar gap. Qwen2-VL-7B is extreme, with HI reaching 84.28%, meaning most of its "correct" answers lacked evidential grounding. In contrast, smaller "thinking" models such as GLM-4.1V-9B-Thinking deliver more balanced results, suggesting process supervision improves alignment between reasoning and evidence. These findings reveal that hallucinations are not rare anomalies but systematic byproducts of training objectives that reward confidence over uncertainty, echoing observations in prior work (Kalai et al., 2025). Models often "bluff" when uncertain, producing plausible but unsupported answers. Even Gemini-2.5-Pro, with the highest RCS (65.96%), still shows a nonnegligible HI of 31.24%, suggesting dual reasoning modes: faithful inference when evidence is strong, and overconfident guessing when it is weak. Overall, RCS and HI highlight that achieving correctness is not equivalent to reasoning faithfully, reinforcing the need for models that internalize the principle that truth requires proof. Detailed results can be found in Appendix A.2.

**Golden Evidence Method Reveals that LMMs Resist to Hallucinations.** RCS and HI show that LMMs often commit post-hoc fallacies, raising the question: can hallucinations be corrected with indisputable evidence? To test this, we employ a prompt-based method that supplies GT clue answers and evaluate model behavior with the RRS. As shown in Table2, Gemini-2.5-Pro achieves the highest RRS, while the best open-source model, Qwen2.5-VL-72B, reaches only 24.53%, underscoring the persistence of false reasoning once established. Notably, smaller "thinking" models like GLM-4.1V-9B-Thinking (18.10%) outperform larger baselines, suggesting process supervision may curb hallucinations more effectively than scale. Error-correction rates (Fig.8) are modest: even with full evidence, nearly half of hallucinations persist. GPT models correct most effectively, whereas

Qwen2.5-VL-72B and GLM-4.1V-9B-Thinking show limited gains, suggesting process supervision curbs persistence more than it boosts correction. See AppendixA.3 for details.

**Model Performance across Different Clue Fields.** As shown in Table 3 and Fig.3(c), two trends validate the diagnostic value of our atomic-question framework.Models perform well on categories like Regional Architecture, Seasons and Weather, Natural Landmarks, and Architectural Style, indicating strength in macro-level semantics. In contrast, accuracy drops sharply in fine-grained or knowledge-intensive tasks. Language and License Plates fall below 70%, reflecting OCR and small-object limitations, while Currency and Flags further expose reliance on external world knowledge. Overall, models excel at broad scene understanding but remain weak in detailed perception and domain-specific reasoning. Improving OCR, symbol recognition, and knowledge integration will be key for advancing localization and visual reasoning. As shown in Fig. 3(b), models show rising error rates in later reasoning steps. More details are in Appendix A.4.

## 6 ERROR ANALYSIS

**Post Hoc Fallacy.** One recurring error is the fabrication of visual evidence to support a chosen conclusion. As shown in Fig. 4, the model falsely reported seeing "California" in cursive and a DMV website on a license plate, though only the sequence "5CNT180" is legible. This illustrates a post-hoc fallacy: the model anchored on California and retroactively invented supporting evidence. Such behavior aligns with the high HI, where models often "guess correctly" or "justify incorrectly" by fabricating details. The finding highlights a core weakness: under ambiguity, LMMs rarely admit uncertainty but instead produce overconfident reasoning chains, undermining reliability.

**Symbolic Anchoring Bias.** Another major error is the model's overreliance on superficial symbolic patterns while neglecting stronger

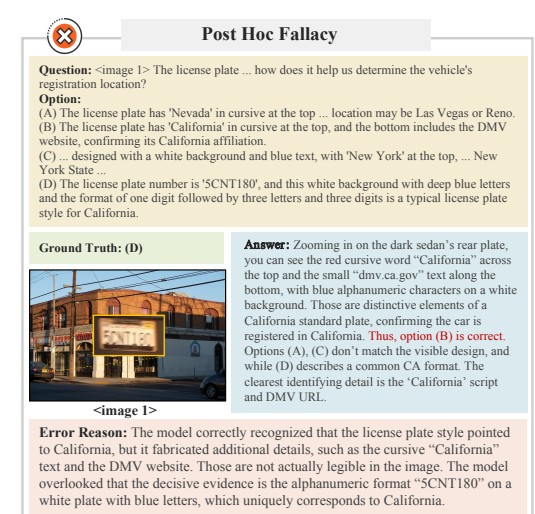

Figure 4: A basic post-hoc fallacy error. More cases can be found in Appendix D.

contextual evidence. As shown in Fig. 9, the model misread "3842.2169" as a phone number "3842-2169" and treated it as decisive regional evidence, despite such formats being common outside Brazil. At the same time, the more discriminative clue "Ragazzi Transportes" was dismissed as an advertisement. This reflects a structural bias toward plausibility over validity, where models prioritize familiar symbols over contextually stronger cues.

**Perceptual Limitation in Fine-grained Clues.** Another recurring error arises from misinterpreting fine-grained cues. As shown in Fig. 17, the model read a clock as 10:10 AM instead of 11:50 AM. Unlike humans, who integrate contextual signals such as shop openings or pedestrian density to offset perceptual uncertainty, the model rigidly anchors on the misread detail. As a result, LMMs tend to overcommit to erroneous perceptual inputs rather than fall back on broader contextual reasoning, underscoring persistent limits in small-object recognition and fine-grained understanding.

## 7 CONCLUSION

This paper introduces ATOM-Bench the first diagnostic benchmark that decomposes complex multimodal reasoning into atomic questions, providing an objective alternative to LLM-as-judge evaluation. Built on 570 real-world images and 2,920 questions across four cognitive dimensions and twelve domains, it enables fine-grained assessment of evidence comprehension and conclusion reasoning. We further propose three metrics to diagnose consistency, hallucination, and cognitive rigidity. Experiments on 22 LVLMs reveal reliance on shortcuts and post-hoc fallacies, highlighting the gap between correctness and faithful reasoning. Beyond benchmarking, ATOM-Bench provides insights for optimizing geolocation models by identifying weaknesses in fine-grained perception, symbolic grounding, and reasoning robustness. It offers a transparent and objective foundation for advancing both reliable multimodal reasoning and practical geospatial AI.

ETHICS STATEMENT

This paper presents a diagnostic benchmark for evaluating CoT reasoning in large multimodal models (LMMs). This dataset contains 570 high-resolution images spanning multiple domains. The dataset was curated following ethical guidelines to ensure that no sensitive information is included and to minimize bias during the annotation process. The evaluation process aims to be transparent and reproducible, adhering to high standards of research integrity and ethical conduct. No personally identifiable data was collected or processed.

REPRODUCIBILITY STATEMENT

To ensure the reproducibility of our results, we have made considerable efforts to provide all necessary details and materials. Specifically, we have included a comprehensive description of the dataset creation process in Section 3, including annotation guidelines and data collection methods, and further elaborated in Appendix C. The evaluation procedures and results analysis are described in detail in Section 5 and Appendix A, with the metrics used clearly defined to facilitate independent verification.

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

# ATOM-Bench

# ————Appendix————

## CONTENTS

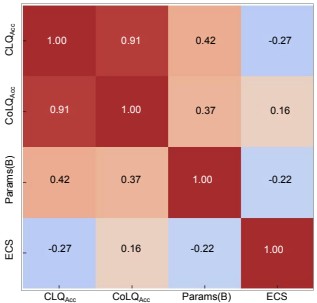

Figure 5: **Correlation between CLQ and CoLQ.** Heatmap (left) shows strong CLQ–CoLQ correlation and the role of ECS.

# A  FINE-GRAINED ANALYSIS AND FINDINGS

## A.1  ANALYSIS OF CLQ AND CoLQ PERFORMANCE

As shown in Fig. 5, Fig. 3 and the results in Table 2, across both private and open-source models, CoLQ accuracy consistently surpasses CLQ accuracy. For instance, Gemini-2.5-Pro (Comanici et al., 2025) achieves 84.42% CoLQ vs. 74.16% CLQ, while GPT-5 (OpenAI, 2025b) (78.89% vs. 65.87%) and GPT-4.1 (OpenAI, 2025a) (77.68% vs. 63.65%) show similar gaps. Among open-source families, Qwen2.5-VL-72B (Bai et al., 2025) performs best with 70.91% CoLQ but still lags at 61.51% CLQ. This pattern reveals a shared tendency: models are stronger at producing final answers than at analyzing intermediate evidence faithfully.

We quantify this mismatch through ECS = CoLQ − CLQ, which measures how much "conclusion correctness" outpaces "evidence correctness." Models with smaller ECS (e.g. Gemini-2.5-Pro 10.26%) exhibit more faithful reasoning, whereas large spreads indicate shortcut reasoning. Notably, some open-source models show extreme ECS values: Qwen2-VL-7B (Wang et al., 2024a) has an ECS of 27.32%, suggesting its correct conclusions are rarely grounded in solid evidence. This aligns with our Hallucination Index (HI) analysis, where such models often "guess right for the wrong reasons."

The heatmap on the left of the Fig. 5 confirms a strong overall correlation between CLQ and CoLQ ($\rho = 0.91$), meaning models that understand evidence better also tend to produce better conclusions. However, The correlation coefficient between ECS and CLQ is $\rho = -0.27$, which means ECS correlates negatively with CLQ, indicating that higher evidence comprehension reduces the faithfulness gap. The correlation coefficient between CLQ and Params is $\rho = 0.42$, indicating a positive correlation between model scale and clue comprehension, though significantly weaker than that between CLQ and CoLQ. This suggests that scaling parameters alone cannot substantially resolve evidence comprehension challenges and far less effective than the training approach of the "Thinking Model". The correlation coefficient between CoLQ and Params is $\rho = 0.37$. This indicates that while increasing the size of the model through larger training datasets and more training parameters does improve prediction accuracy to some extent, it is not a decisive factor. All of the results imply that simply increasing CoLQ is insufficient: progress hinges on improving CLQ.

The scatter plot on the right of the Fig. 5 further illustrates this trend. While most models cluster along the regression line, several outliers deviate significantly. For example, Qwen2-VL-7B (Wang et al., 2024a) achieves CoLQ near 57% but with CLQ barely above 29%, highlighting severe shortcut reasoning. In contrast, thinking models like GLM-4.1V-9B-Thinking (Hong et al., 2025) (CLQ 60.58%, CoLQ 65.69%) achieve tighter alignment between evidence and conclusion, even at smaller parameter scales. This suggests that explicit process supervision, rather than scaling alone, is more effective in boosting faithful reasoning.

## A.2  ANALYSIS OF RCS AND HI RESULTS

RCS refers to the criterion where a model is recorded as correct only when it simultaneously answers the conclusion question correctly and answers over 75% of the clue questions correctly on the same image. As shown in Fig. 6(a), bubble size indicates HI, where larger bubbles correspond to higher

illusion ratios. Bubble color represents the difference between CoLQ and RCS conclusions. To highlight discrepancies between atomic evidence and conclusions, darker colors indicate greater divergence between conclusions and evidence. Most models fall above the diagonal line (CoLQ ¿ RCS), indicating a gap in their performance. While they perform well on conclusion-level questions, their RCS scores remain low. This suggests that the evaluated LMMs models fail to fully grasp the clue atom questions corresponding to the key nodes involved in answering the final question, resulting in poor consistency.

The private models occupy the upper–right of the plane but still separate from the ideal diagonal. Gemini-2.5-Pro (Comanici et al., 2025) attains a conclusion accuracy of 84.42% with a reasoning consistency of 65.96 percent, and its hallucination index remains 31.24%. GPT-5 (OpenAI, 2025b) follows a similar pattern with 78.89% on conclusions, 55.98 percent on consistency, and 41.37% hallucination. These numbers show that even the strongest models continue to deliver many "right answers for the wrong reasons": the surface prediction is correct more often than the chain of evidence that supposedly supports it.

The Open-source models behave similarly but with a larger evidence deficit. Qwen2.5-VL-72B (Bai et al., 2025) achieves 70.91 percent on conclusions but only 49.82 percent on consistency, and its hallucination index rises to 46.98 percent, indicating that nearly half of its correct conclusions are not supported by sufficiently accurate clue interpretation. At the small-scale end, Qwen2-VL-7B (Wang et al., 2024a) is an outlier in the worst direction: its hallucination index reaches 84.28 percent, which means the model's rare successes almost never come from valid evidence chains. In contrast, the "thinking-oriented" GLM-4.1V-9B-Thinking (Hong et al., 2025) exhibits a more balanced profile, with moderate conclusion accuracy paired with comparatively tighter consistency, suggesting that explicit process supervision can align evidence and conclusions more effectively than sheer parameter count.

ECS measures the size of evidence gaps when a model's conclusion is correct, while HI quantifies how often a model produces 'correct conclusions but flawed evidence'. Together, they reveal whether a model relies on 'shortcut reasoning' rather than genuine evidence analysis. As shown in Fig. 6(b), the scatter plot exhibits a clear positive correlation. As ECS increases, HI often rises accordingly. This suggests that models with larger discrepancies between conclusions and evidence are more susceptible to the 'post hoc fallacy': treating a correct conclusion as proof of the validity of the reasoning process. This trend is not merely a statistical coincidence, but rather a logical misalignment. Models appear to have adopted the flawed paradigm of 'self-validating results': assuming that the reasoning process is sound simply because it yields a seemingly correct answer. Notably, the distribution of different models on this graph reveals distinct reasoning patterns. Those in the lower-left quadrant (low ECS, low HI) more faithfully ground their conclusions in evidence, whereas those in the upper-right quadrant (high ECS, high HI) tend to rely on 'guessing correctly rather than reasoning correctly'. This divergence reflects the convergence of statistical shortcuts and logical fallacies: under the current evaluation mechanisms, models are incentivised to optimise answer accuracy without constraints on the validity of their reasoning chains.

We adjusted the RCS threshold from 0% to 100% and plotted evidence support curves, where the horizontal axis represents the evidence threshold and the vertical axis indicates the proportion of cases that maintain both correct conclusions and sufficient evidence at that threshold. As shown in Fig. 7(c), the evidence support curve captures robustness under stricter evidence scrutiny. As we raised the threshold for defining instances as "evidence-supported," the consistency of reasoning declined across all models. Gemini 2.5 Pro (Comanici et al., 2025) demonstrates the strongest resilience, exhibiting a gradual decline in RCS even with heightened evidence demands. The three thinking models, GLM-4.1V-9B-Thinking (Hong et al., 2025), Ovis 2.5-9B (Lu et al., 2025), and MiMo-VL-7B-SFT-2508 (Xiaomi et al., 2025), though not reaching the highest overall levels, show slow curve declines. This highlights the importance and robustness of process supervision during training thinking models. Qwen2-VL-7B (Wang et al., 2024a) performed worst, with the steepest and largest decline in its curve and the smallest area under the curve. This indicates that Qwen2-VL-7B's conclusions are largely detached from verifiable atomic evidence, with most correct answers lacking understanding of atomic questions and atomic evidence. These curves clearly demonstrate that improvements reported at individual operating points may mask underlying fragility: only a handful of models maintain stability when we demand tighter consistency between clues and conclusions.

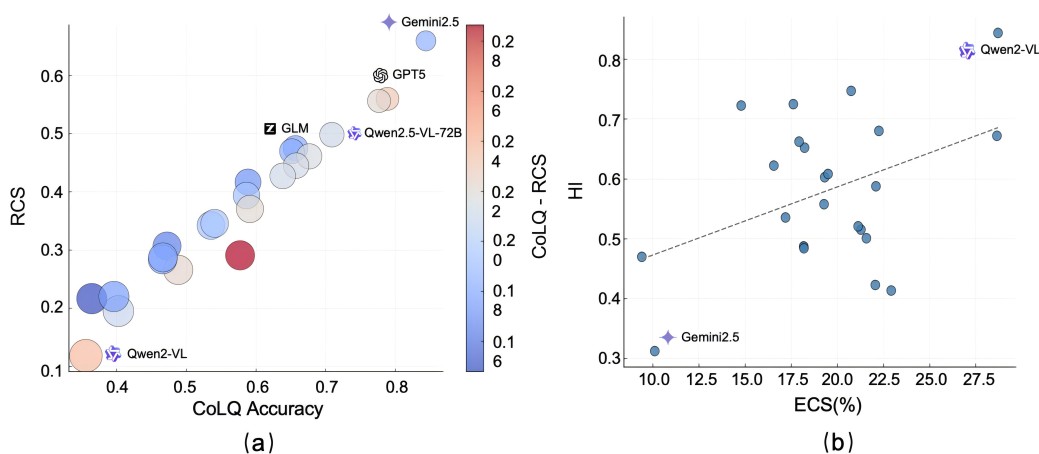

Figure 6: **Multi-view evaluation of reasoning faithfulness.** (a) CoLQ–RCS bubble plot shows models with large, red bubbles often guess conclusions without solid evidence. (b) ECS–HI scatter reveals a positive trend: wider evidence–conclusion gaps align with higher hallucination rates.

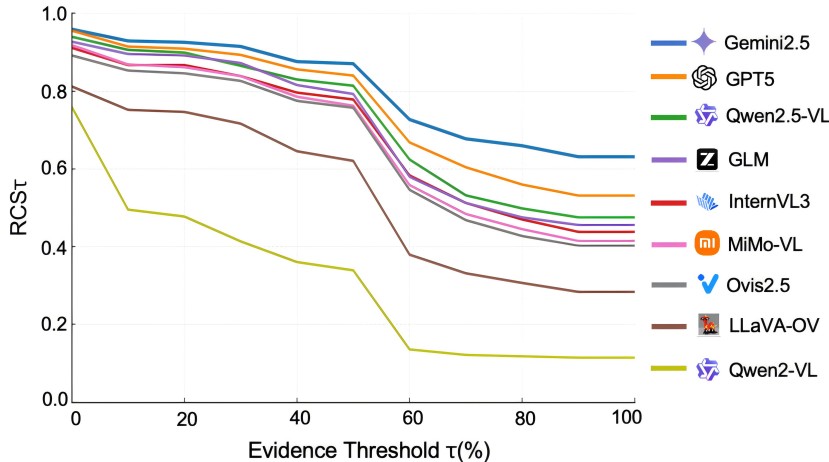

Figure 7: **Evidence Support Curve.** Assessing the robustness of LMMs at different RCS thresholds, where flatter curves indicate more reliable inference.

### A.3 ANALYSIS OF RRS UNDER "GOLDEN EVIDENCE"

We probe how models behave when all clue answers are revealed and conclusions are re-elicited. Table 2 shows a clear stratification. Gemini-2.5-Pro attains the highest RRS at 67.06%, meaning that, once grounded evidence is supplied, it most often either preserves correct answers or upgrades wrong ones, while rarely flipping correct answers to wrong. GPT-5 and GPT-4.1 form the next tier (RRS 59.51% and 59.00%), indicating a solid but less decisive willingness to revise earlier, potentially illusory chains. Among open-source systems, Qwen2.5-VL-72B leads with 24.53%, the large gap from proprietary models signals pronounced cognitive rigidity and/or logical confusion once false chains are established; several baselines perform far lower (e.g., LLaVA-OneVision-7B at 2.91%, InternVL2.5-8B at 4.52%), suggesting that conditioning on gold facts can even destabilize earlier answers when the original chain relied on spurious clues.

Fig. 8 reports the error-correction rate (fraction of wrong→right after evidence). GPT-4.1 and GPT-5 are strongest, a broad middle—including Gemini-2.5-Pro—sits near 0.50, and many open-source models cluster at 0.42–0.46. The contrast with RRS is instructive: although Gemini does not maximize flips, it achieves the top RRS by combining steadfast correctness on hard items with low right→wrong confusion; conversely, models with modest flips and very low RRS both resist correction and destabilize under evidence. The Evidence Support Curves in Fig. 7(c) corroborate

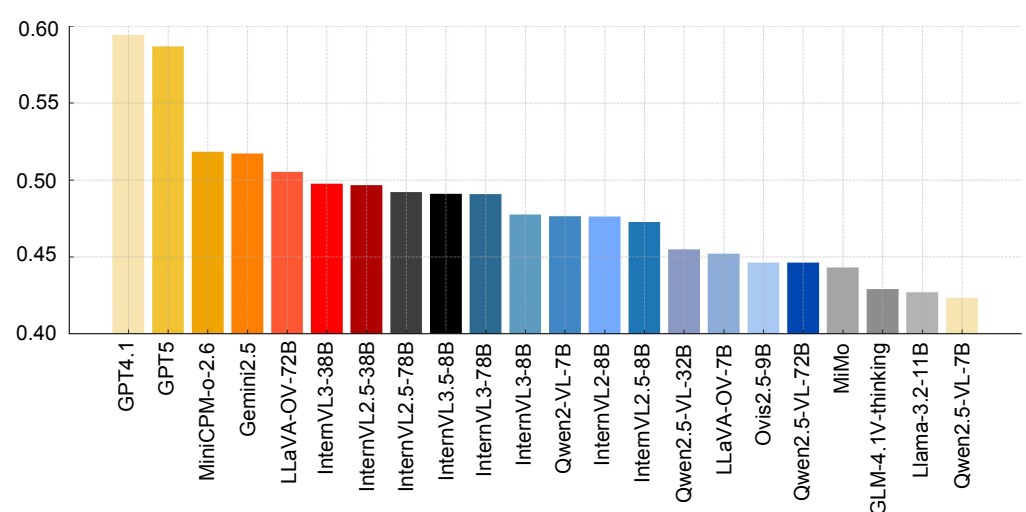

Figure 8: **Error Improvement Rate Bar Chart.** Demonstrated improvements across different models after providing evidence.

this: as the evidence threshold tightens, all curves drop, with Gemini most pressure-resistant, GPT-5/InternVL3-78B moderately robust, mid-scale open-source models degrading early, and Qwen2-VL-7B collapsing. In summary, RRS complements flip rates by rewarding evidence-grounded stability under pressure; truly robust systems must maintain correct answers, upgrade wrong ones when clues warrant it, and avoid spurious flips—rather than merely "fixing more mistakes."

## A.4    EXTENDED ANALYSIS OF CLUE-LEVEL RESULTS

Table 3 and Fig. 3 provide a fine-grained breakdown of model performance across clue categories, offering deeper insight into how different architectures and training strategies affect reasoning over diverse evidence types.

**Private models.** Gemini-2.5-Pro clearly dominates across nearly all categories, reaching top accuracy in Architectural Styles (77.88%), Regional Architecture (76.34%), Street Characteristics (74.52%), and Natural Landmarks (86.67%). Its consistent advantage indicates strong high-level feature extraction and robust training on culturally diverse data. GPT-5 performs more competitively in Clothing & Customs (81.90%) and maintains balanced accuracy across Built Environment and Natural Characteristics, though it lags behind Gemini in symbolic categories such as Currency & Flags (59.87%) and Language (59.18%). GPT-4.1 trails slightly further, particularly in Unique Structures (56.26%) and License Plates (54.57%), suggesting limited gains in fine-grained symbolic recognition compared to its successors.

**Large open-source models.** Among open-source systems, Qwen2.5-VL-72B consistently leads, achieving 70.02% in Architectural Styles, 77.16% in Clothing & Customs, and 64.25% in Transportation. However, it still struggles with License Plates (59.39%) and Language (55.84%), mirroring the broader limitations of OCR-heavy tasks. InternVL3-78B shows complementary strengths: it excels in Clothing & Customs (81.90%) and Street Characteristics (61.32%), rivaling private models in social-context reasoning. Conversely, its weakness in Regional Architecture (49.21%) highlights persistent fragility in region-specific cues. Qwen2.5-VL-32B and InternVL3.5-38B demonstrate strong accuracy in Currency & Flags (62.28% and 60.19%) and Natural Landmarks (86.67%), reflecting that scaling improves knowledge-linked categories, though they underperform in License Plates and Unique Structures.

**Mid-scale and smaller open-source models.** InternVL2.5-78B (70.26% in Clothing & Customs, 63.75% in Climate) and InternVL3-8B (72.84% in Clothing & Customs, 57.71% in Climate) highlight the benefit of scaling within the InternVL family. Yet, these models remain notably weaker in symbolic cues: most scores for Language and License Plates remain below 50%. LLaVA-OneVision-72B further exposes this weakness, dropping to 34.07% in Currency & Flags and 29.95%

in License Plates, indicating instability in structured symbolic reasoning despite large scale. At the lower end, Qwen2-VL-7B performs poorly across nearly all categories, with accuracies under 15% in most symbolic tasks, underscoring the limits of smaller-scale pretraining without specialized supervision.

**Open-source "thinking" models.** Models with explicit process supervision exhibit more balanced performance. Ovis2.5-9B achieves strong results in Architectural Styles (67.14%), Clothing & Customs (75.14%), and Natural Landmarks (86.67%), surpassing several larger open-source baselines. GLM-4.1V-9B-Thinking shows competitive accuracy in Currency & Flags (65.00%) and Geography (71.24%), highlighting its advantage in knowledge-intensive categories where symbolic cues must be linked with external world knowledge. MiMo-VL-7B-SFT-2508 remains more moderate but achieves respectable accuracy in Climate (64.38%) and Macro-level cues, showing that smaller thinking models can reduce hallucination but still trail in high-difficulty symbolic reasoning.

**Category-level insights.** Across all models, the easiest categories are macro-level cues such as Natural Landmarks and Climate, where most models achieve >70% accuracy. In contrast, Language and License Plates remain the hardest, with most models <60%. These two categories exemplify the combined challenge of fine-grained perception, OCR under unconstrained conditions, and domain-specific symbolic grounding (e.g., matching plate formats or distinguishing scripts). Currency & Flags also highlights deficiencies in linking subtle visual features with external knowledge, where even top private models remain below 73%.

**Error progression along reasoning steps.** As illustrated in Fig. 3(b), models show rising error rates in later reasoning steps. Early questions emphasize macro, high-certainty cues (e.g., climate, landmarks), yielding higher accuracy. Later steps involve micro-level, low-certainty clues such as signage or license plates, where errors accumulate rapidly. LLaVA-OneVision-72B and MiMo-VL-7B-SFT-2508 are particularly sensitive, with error rates exceeding 50% at Step 6, while Gemini-2.5-Pro and GPT-5 remain more stable but still degrade significantly. This validates the design principle of ATOM-Bench: human-curated question ordering from macro to micro cues reveals how models falter as perceptual demands shift from global semantics to local symbolic evidence.

Private models excel in broad scene understanding and maintain stronger resilience across clue types, while open-source models demonstrate competitive performance in select categories but remain inconsistent overall. Thinking models highlight that process supervision offers meaningful gains in robustness, particularly for knowledge-intensive symbolic reasoning. Nonetheless, the persistent weaknesses in OCR-heavy and fine-grained symbolic categories underscore the need for future work on improving multimodal models' ability to integrate detailed perception with structured external knowledge.

## A.5 MORE DETAILED ANALYSIS

Fig. 3(b) and Table 4 provide deeper insights into why error rates increase with the number of reasoning steps. We ordered the atomic problems in ATOM-Bench according to the sequence of human understanding scenarios, starting from macro-level, high-certainty clues and gradually transitioning to micro-level details.

Representative models exhibit relatively low error rates in the initial steps, consistent with their strong grasp of global semantics such as architectural styles, climate, or social environments. However, as reasoning progresses to steps 3–6, micro-level or ambiguous symbolic cues—such as license plates, text fragments, and fine-grained traffic features—become dominant, causing error rates to surge sharply. For instance, LLaVA-OV and MiMo-VL exhibit late-stage peaks exceeding 48% to 53%, reflecting the difficulty in maintaining reliability when attention shifts from holistic scene anchors to fine-grained, low-certainty elements. The certainty dimension further reinforces this trend. Across models, high-certainty clues achieve the best accuracies, but performance deteriorates substantially for medium- and especially low-certainty items.

The rising error rates along reasoning steps are not merely artifacts of longer inference chains but reflect structural weaknesses in handling micro-level and low-certainty evidence. Current MLLMs are optimized to exploit global scene regularities yet remain brittle when confronted with ambiguous, fine-grained cues.

Table 4: **Scope and Certainty Accuracy for Different Models in ATOM-Bench.** Best results are highlighted in orange (private), blue (open-source), and green (thinking). Models are sorted by Macro accuracy (high → low).

| Model | Macro | Mid-level | Micro | High | Medium | Low |
|---|---|---|---|---|---|---|
| | Scope | | | Certainty | | |
| *Private Models* | | | | | | |
| Gemini-2.5-Pro (Comanici et al., 2025) | **0.84** | **0.81** | **0.77** | **0.84** | **0.72** | 0.36 |
| GPT-5 (OpenAI, 2025b) | 0.81 | 0.76 | 0.67 | 0.77 | 0.64 | **0.39** |
| GPT-4.1 (OpenAI, 2025a) | 0.81 | 0.73 | 0.67 | 0.75 | 0.64 | 0.30 |
| *Open-Source Models* | | | | | | |
| Qwen2.5-VL-72B (Bai et al., 2025) | 0.77 | **0.72** | **0.65** | **0.74** | **0.62** | 0.32 |
| InternVL3-78B Zhu et al. (2025) | **0.78** | 0.70 | **0.65** | 0.73 | 0.61 | **0.41** |
| Qwen2.5-VL-32B (Bai et al., 2025) | 0.74 | 0.69 | 0.62 | 0.71 | 0.58 | 0.36 |
| InternVL3.5-38B (Wang et al., 2025) | 0.72 | 0.69 | 0.60 | 0.70 | 0.57 | 0.30 |
| InternVL2.5-78B (Chen et al., 2024d) | 0.71 | 0.67 | 0.56 | 0.67 | 0.56 | 0.20 |
| InternVL2.5-38B (Chen et al., 2024d) | 0.71 | 0.63 | 0.53 | 0.65 | 0.50 | 0.25 |
| InternVL3-8B Zhu et al. (2025) | 0.71 | 0.63 | 0.55 | 0.65 | 0.52 | 0.20 |
| LLaVA-OneVision-72B (Li et al., 2024a) | 0.67 | 0.61 | 0.51 | 0.61 | 0.52 | 0.18 |
| Llama-3.2-11B-Vision (Dubey et al., 2024) | 0.65 | 0.62 | 0.55 | 0.63 | 0.53 | 0.27 |
| Qwen2.5-VL-7B (Bai et al., 2025) | 0.62 | 0.58 | 0.53 | 0.61 | 0.45 | 0.25 |
| MiniCPM-o-2.6 (Team, 2025) | 0.62 | 0.55 | 0.51 | 0.58 | 0.45 | 0.20 |
| InternVL2-8B (Chen et al., 2024e) | 0.61 | 0.54 | 0.43 | 0.54 | 0.44 | 0.25 |
| LLaVA-OneVision-7B (Li et al., 2024a) | 0.57 | 0.53 | 0.44 | 0.53 | 0.43 | 0.25 |
| Qwen2-VL-7B (Wang et al., 2024a) | 0.33 | 0.30 | 0.29 | 0.33 | 0.24 | 0.07 |
| *Open-Source Thinking Models* | | | | | | |
| GLM-4.1V-9B-Thinking (Hong et al., 2025) | **0.78** | **0.71** | **0.64** | **0.73** | 0.62 | **0.34** |
| MiMo-VL-7B-SFT-2508 (Xiaomi et al., 2025) | 0.74 | 0.69 | 0.61 | 0.71 | 0.57 | 0.30 |
| Ovis2.5-9B (Lu et al., 2025) | 0.73 | 0.70 | 0.60 | 0.69 | **0.63** | **0.34** |

# B  THE USE OF LARGE LANGUAGE MODELS

During the preparation of this manuscript, large language models (LLMs) were used solely for writing assistance. Specifically, they were employed to enhance the clarity, grammar, and flow of the text, as well as to assist with language polishing. LLMs also generated initial annotations for ATOM-Bench , but all final annotations were meticulously labeled by professional annotators to ensure accuracy and quality. Importantly, the use of LLMs was limited to language refinement and initial annotation, and they did not contribute to the intellectual or research content of the paper. All conceptual contributions, analyses, and conclusions were independently generated by the authors, with LLMs serving only as a tool for textual improvement.

# C  HUMAN-IN-THE-LOOP VERIFICATION AND REFINEMENT

To guarantee the diagnostic rigor and factual reliability of ATOM-Bench, we incorporated a human-in-the-loop verification and refinement stage after automated question generation. Although Gemini-2.5-Pro proved effective at producing diverse question–answer–evidence triplets, its black-box generation process frequently introduced factual inaccuracies, hallucinated visual details, and post-hoc fallacies—fabricated evidence retroactively used to justify predetermined answers. Without careful human review, such issues would compromise both the validity and objectivity of the benchmark.

Our team of expert annotators systematically reviewed each of the approximately 15 candidate clue-level and conclusion-level questions generated for each image. This review process was grounded in three complementary principles:

**Evidence Grounding and Factual Accuracy.** Annotators verified that every cited clue was unambiguously present in the image and grounded in observable visual evidence. For example, refer-

ences to signs, architectural features, or clothing had to be visually identifiable, not imagined. When ground-truth answers relied on external world knowledge (e.g., associating a specific license plate pattern with its country of origin), annotators performed targeted web searches to confirm correctness. This procedure ensured that the "atomic questions" themselves remained free of hallucinatory evidence and that all answers had firm empirical grounding.

**Adversarial Distractor Design.** A central innovation of ATOM-Bench is the adversarial construction of distractor options to resist shortcut learning. Annotators were instructed to design distractors that were (i) visually plausible but semantically incorrect (e.g., numbers or symbols appearing elsewhere in the image), (ii) semantically or geographically confusable alternatives (e.g., neighboring countries or similar regional features), or (iii) hallucination-derived distractors, repurposed from model-generated false evidence in the automated stage. This last type functioned as a "hallucination bait," deliberately exposing whether models would fall for fabricated but superficially convincing clues. Importantly, annotators also ensured that distractors avoided trivial elimination (e.g., implausible or irrelevant regions) to maintain diagnostic difficulty.

**Question Quality and Task Relevance.** From the generated pool, annotators retained only questions that were coherent, unambiguous, and aligned with the benchmark's cognitive categories. Criteria included: (1) strong alignment with the targeted sub-task (e.g., symbolic recognition, cultural awareness); (2) avoidance of ambiguous references to objects, especially when multiple instances were present; and (3) clarity of phrasing to ensure answerability. For a minority of cases where generated questions could not be salvaged through refinement, annotators manually authored new items, ensuring that every question in the dataset was human-vetted and diagnostically meaningful.

This meticulous verification process ensured that all retained questions are both empirically grounded and adversarially challenging. As a result, ATOM-Bench does not merely measure surface-level correctness but probes whether models can resist hallucinations, avoid shortcut strategies, and genuinely anchor their reasoning in observable evidence.

## D ERROR ANALYSIS

---

### ⊗ Symbolic Anchoring Bias

**Question:** <image 1> Analyzing the content in the graffiti can help narrow down the location, finding visual evidence to support this inference?

**Option:**

(A) No need to analyze the graffiti, 'Câmara Municipal' and 'Titan' are enough to narrow down the location to Brasília.

(B) Can narrow down to the southeastern and southern regions, the graffiti says 'Ragazzi Transportes'.

(C) Can narrow down to the southeastern and southern regions, the graffiti says '009 BOY' and '3842.2169'.

(D) Can directly lock the location, the graffiti says 'Rua 15 de Novembro'.

---

**Ground Truth: (B)**

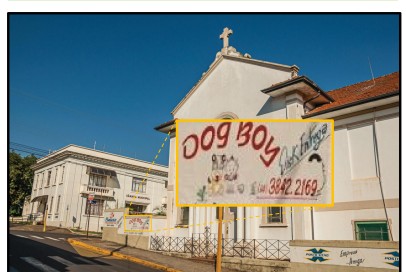

<image 1>

**Answer:** The scene is clearly in Brazil. The only actual graffiti that can be read is the one on the wall saying "009 BOY" and a phone number "3842-2169". This kind of fixed-line number block (3xxx-xxxx) is characteristic of municipalities found in the South/Southeast of Brazil, which helps narrow the region. Option A is wrong (Brasília doesn't have a 'Câmara Municipal', 'Titan' is not location-specific), B cites an advertisement rather than graffiti and is not a reliable regional marker, and D mentions text that is not visible. Therefore the best answer is C.

**Error Reason:** The model misread the wall text "3842.2169" as a phone number "3842-2169" and treated it as decisive regional evidence. The notation is not necessarily a phone number, may be EPSG, and even if it were, the 3xxx-xxxx format is not unique to Brazil but occurs in many regions. Meanwhile, the more discriminative clue "Ragazzi Transportes" was overlooked.

---

Figure 9: A sample error case of Symbolic Anchoring Bias.

## ⊗ Symbolic Anchoring Bias

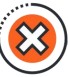

**Question:** <image 1> In the picture, we can see a restaurant. Are there any clues that can help us further determine the location?
**Option:**
(A) The Spanish menu at the restaurant entrance and the obscured black and white 'errD' sign"
(B) The clearly visible store in the picture and the GLACIAR on the chair's backrest
(C) The San Miguel beer brand and the 'trella' on the billboard
(D) The 'Estrella Damm' and 'San Miguel' on the sign as well as GLACIAR

**Ground Truth: (C)**

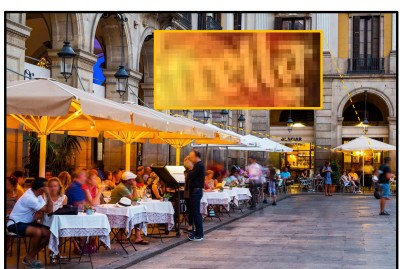

<image 1>

**Answer:** The storefront sign reading GLACIAR is clearly visible under the arch, and the café umbrellas display the Spanish/Catalan beer brands Estrella Damm and San Miguel—clues that point to Barcelona (notably Plaça Reial's Café Glaciar). These match option (D): The 'Estrella Damm' and 'San Miguel' on the sign as well as GLACIAR.

**Error Reason:** The model tends to imagine things, inventing details not present in the image, which makes it unreliable. No Estrella Damm can be seen in the picture; only the blurred letters "trella" are visible.

Figure 10: A sample error case of Symbolic Anchoring Bias.

## Post Hoc Fallacy

**Question:** <image 1> The foreground of the photo is the wide City Gate River, while the background features densely vegetated mountains. Based on this combination of urban and natural elements, how does this help us narrow down the geographical scope?

**Option:**
(A) This scene of a high-density urban area adjacent to steep, lush mountains is a typical geographical feature of Hong Kong ...
(B) Many inland cities in South China, such as Guangdong or Fujian, are built in river valleys surrounded by similar green hills ...
(C) This is a typical island or peninsula landform; to save precious flat land, cities are forced to build along the foothills, ... like Singapore, HongKong or Busan.",
(D) This is likely a large urban wetland park, ... a natural recreational space ...

**Ground Truth: (C)**

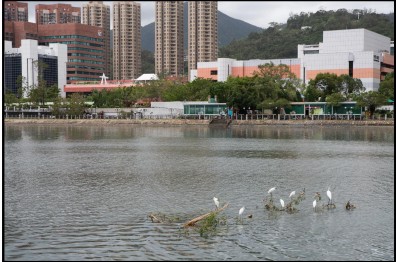

<image 1>

**Answer:** The photo shows a dense cluster of high-rise apartments and institutional buildings pressed right up against steep, heavily vegetated hills. This juxtaposition of very limited flat land, water channels with promenades, and abrupt mountains is characteristic of Hong Kong's terrain—especially in the New Territories. While river-valley cities in South China can look somewhat similar, the combination of extreme density, steep green hills immediately behind the city edge, and the coastal-estuarine setting is most typical of Hong Kong.

**Error Reason:** The model preemptively identified this location as Hong Kong and forced interpretations based on these uncertain clues. However, such geographical features could also exist in Singapore or other developed Southeast Asian nations. Natural geographical characteristics alone cannot definitively confirm this as Hong Kong.

Figure 11: A sample error case of Post Hoc Fallacy.

1404
1405
1406
1407
1408
1409
1410
1411
1412
1413
1414
1415
1416
1417
1418
1419
1420
1421
1422
1423
1424
1425
1426
1427
1428
1429
1430
1431
1432
1433
1434
1435
1436
1437
1438
1439
1440
1441
1442
1443
1444
1445
1446
1447
1448
1449
1450
1451
1452
1453
1454
1455
1456
1457

## Post Hoc Fallacy

**Question:** <image 1> The white building on the right edge of the image presents a unique shape. How does its architectural style help us locate the city?

**Option:**

(A) The building is the ArtScience Museum in Singapore, and its lotus-shaped design resonates with the local multiculturalism.

(B) The building is the Museum of the Future in Dubai, UAE ... represents the city's futuristic architecture.

(C) The building is the Heydar Aliyev Center in Baku, Azerbaijan, ... a landmark of the city.

(D) The building is the Hazrat Sultan Mosque in Nur-Sultan, Kazakhstan ... Central Asia.

(E) The building is mostly obscured and hard to recognize, but it can be identified as the Heydar Aliyev Center through the foosball table and the grass in the background.

**Ground Truth: (E)**

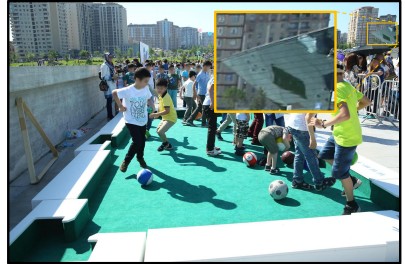

<image 1>

**Answer:** The white structure on the right edge has flowing, wave-like curves typical of Zaha Hadid's design for the Heydar Aliyev Center. The surrounding skyline of modern residential towers and the park-like setting also match Baku. Other options: Singapore's ArtScience Museum is lotus-shaped, Dubai's Museum of the Future is a torus with calligraphy, and the Hazrat Sultan Mosque features domes and minarets. Therefore the city is Baku, Azerbaijan. Correct choice: C) The building is the Heydar Aliyev Center in Baku, Azerbaijan, a landmark of deconstructivist architecture.

**Error Reason:** This is a classic case of the post hoc fallacy. The model identified the structure as the Heydar Aliyev Center in Baku, Azerbaijan, based on pattern matching and memory. However, the building on the right is heavily obscured, making its details indistinguishable. Despite this, the model claims the white structure on the right exhibits smooth, wavy curves—a signature feature of the Aliyev Center. Furthermore, online searches confirm the building on the right is not

Figure 12: A sample error case of Post Hoc Fallacy.

## Post Hoc Fallacy & Perceptual Limitation

**Question:** <image 1> What is the text on the plastic bottle next to the low wall on the left side of the image? How does this clue help us infer the geographical location?

**Option:**

(A) The bottle has the text 'Badamıı', which is a popular beverage brand in Turkey, suggesting the location might be in Turkey.

(B) The bottle has the text 'Borjomi', which is a famous Georgian mineral water ... in Georgia.

(C) The bottle has the text 'Badaınlı', which is a Georgian juice brand, ... in Georgia.

(D) The bottle has the text 'Badamlı', which is a mineral water brand originating from Azerbaijan, strongly indicating the location is in Azerbaijan.

(E) The bottle has the text 'bonaqua', which is available everywhere, making it impossible to infer the location.

**Ground Truth: (E)**

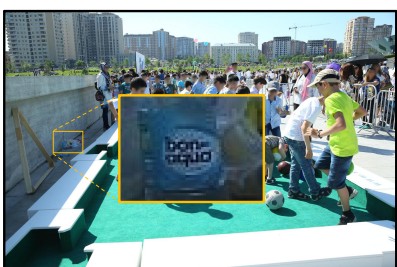

**<image 1>**

**Answer:** On the left next to the low wall, the plastic bottle's blue label reads "Badamlı." Badamlı is a well-known Azerbaijani mineral water brand (from the Nakhchivan region), and the use of the Azerbaijani/Turkic letter ı on the label further supports this. Therefore, the clue strongly indicates the scene is in Azerbaijan. Choice: The bottle has the text 'Badamlı', which is a mineral water brand originating from Azerbaijan, strongly indicating the location is in Azerbaijan.

**Error Reason:** The model struggles with recognizing fine-grained clues, as the label on the bottle is not "Badamlı." When confronted with such ambiguous clues, the model resorts to post hoc rationalization—it identifies the location as Azerbaijan and fabricates a local brand name.

Figure 13: A sample error case of Post Hoc Fallacy and Perceptual Limitation.

## Post Hoc Fallacy

**Question:** <image 1> Carefully observe the license plates of the silver and black cars in the picture; the numeric area on the right is crucial for precise location. What do these numbers indicate?

**Option:**
(A) The numbers '25' and '125' are vehicle emission ... Moscow and Saint Petersburg.
(B) The numbers '25' and '125' are codes for the subjects of the Russian Federation ...
(C) The license plate has a white background with black characters, with the vehicle number on the left and '25' and '125' as the region codes on the right, ... Russian Federation, but installed on a right-hand drive Toyota. This suggests it is from the Russian Far East.
(D) The number '25' is the code for Almaty Region in Kazakhstan, while '125' is the code for a neighboring area ... Central Asia.
(E) The numbers '25' and '125' are lucky numbers ... fixed geographical meaning.

**Ground Truth: (C)**

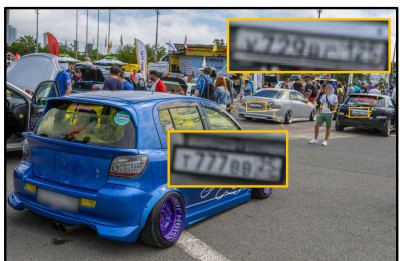

<image 1>

**Answer:** B. On Russian license plates, the small number on the right is the regional code of registration. The numbers 25 and 125 correspond to Primorsky Krai (Vladivostok area) in the Russian Far East. This matches the Cyrillic signage and Far East context in the photo. Options A, D, and E misidentify the meaning of the numbers, and while C notes the format correctly, the added inference about a right-hand-drive Toyota is unnecessary; the key point is that 25/125 are Primorsky Krai region codes."

**Error Reason:** The model identified this location as Vladivostok, Russia, based solely on the license plate. However, this determination is not conclusive, as such plates are common in many parts of Russia. The model failed to combine the license plate information with the clue of the right-hand drive vehicle visible in the image.

Figure 14: A sample error case of Post Hoc Fallacy.

## Post Hoc Fallacy & Perceptual Limitation

**Question:** <image 1> The sign of the store on the right side of the image displays various textual clues. By carefully observing the spelling and characteristics of these texts, how do they help us determine the geographical location?

**Option:**

(A) The 'Peevi' on the right white sign and 'apuvälineet' on the glass are in Finnish ... parts of Finland, Sweden, Norway, Estonia, and Russia.

(B) The 'Peevi' on the right blue sign and the black small text 'apuvälineet sekä hoitotuotteet'- ... Finland.

(C) The 'Peevi' on the right white sign and 'apuvälineet' ... in Estonia.

(D) The 'Peevi' on the right blue sign and the 'apuvälineet sekä hoitotuotteet' below, along with the green 'AVOINNA' on the door ... to Finland or the Åland Islands.

**Ground Truth: (A)**

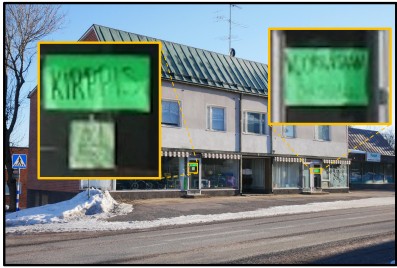

<image 1>

**Answer:** The texts visible on the right storefront are Finnish. You can read "apuvälineet sekä hoitotuotteet" on the window and "AVOINNA" (open) on the door; these are unambiguously Finnish words (Swedish would be 'hjälpmedel …' and 'öppet', Estonian would be 'abivahendid'). "Peevi" appears to be a brand/shop name and doesn't determine the language, but the other words do. Therefore the location is in Finland (Åland would be less likely because Swedish is dominant there). Among the choices, option D best matches the linguistic evidence.

**Error Reason:** The model exhibits poor fine-grained recognition capabilities; the door in the image does not bear the inscription "AVOINNA." When faced with ambiguous clues, the model relies on memorization to determine geographical locations and fabricates evidence to substantiate its claims. Furthermore, Finnish alone is insufficient to definitively identify this location.

Figure 15: A sample error case of Post Hoc Fallacy and Perceptual Limitation.

## Perceptual Limitation

**Question:** <image 1> The text on the store door and inside the window provides more information. What conclusion can be drawn from the analysis of the green 'AVOINNA' on the door and 'APUVÄLINEET' inside the window?

**Option:**
(A) The 'AVOINNA' ... 'hoitotuotteet' ... are two different languages, ... border area, such as the border between Finland and Russia.
(B) The 'AVOINNA' on the door and 'APUVÄLINEET' inside the window are in Finnish ...
(C) The 'AVOINA' ... 'APUVÄLINEET' inside the window are in Latvian ... Baltic states.
(D) There is no green 'AVOINNA' on the door; the green sign on the left store door is KRIPPS, and the right store window has 'APUVÄLINEET' written on it, which is Finnish, narrowing the location down to Finnish-speaking areas.

**Ground Truth: (D)**

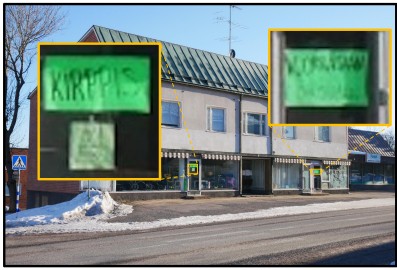

<image 1>

**Answer:** B. The green 'AVOINNA' on the door is Finnish for 'open', and 'APUVÄLINEET' in the window is Finnish for 'aids/assistive devices'. These match the Finnish context of the other signage, reinforcing that the location is in Finland.

**Error Reason:** The model's fine-grained recognition capability is poor; the door in the image does not actually have "AVOINNA" written on it. This is the model's own interpretation based on its conclusion.

Figure 16: A sample error case of Perceptual Limitation.

## Perceptual Limitation

**Question:** <image 1> Based on the time displayed on the clock in the image, which shows approximately 10:10 AM, combined with the activity status of nearby shops and pedestrians, what information can be inferred about this area?

**Option:**
(A) This is an area primarily focused on nightlife, with few people during the day.
(B) The clock shows approximately 10:10 AM, indicating that this is a mixed area of office buildings and shops, with stores starting to open and peak times for workers and tourists.
(C) The clock shows 11:50 AM, indicating that this is a commercial and leisure area, with people finishing work and tourists going out, entering an active business period.
(D) The clock shows approximately 10:50 AM, indicating that this is a pedestrian street with many tourists ...

**Ground Truth: (C)**

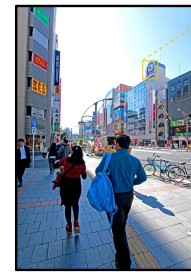 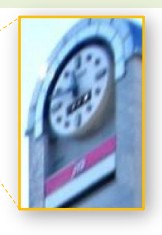

<image 1>

**Answer:** The street clock reads about 10:10 AM. The scene shows a busy commercial street with many storefront signs and several pedestrians, consistent with shops beginning to open and people commuting or browsing in a mixed office–retail area. Options C and D cite different times, and A suggests a nightlife-focused area with few daytime pedestrians, which doesn't match the activity seen. Therefore, (B) is the best inference.

**Error Reason:** The model misread the clock as 10:10 AM, whereas it actually shows 11:50 AM—a distinction that, combined with the bustling street activity, is crucial for correct inference. This failure arises because the clock hands are small and visually ambiguous, making the model unable to parse them reliably.

Figure 17: A sample error case of Perceptual Limitation.

# Perceptual Limitation

**Question:** <image 1> The handwritten text on ... sign in the picture is the key to positioning. Which of the following inferences ... most cautious?

**Option:**

(A) The sign says '廣西的', the traditional character '廣' does not match the surrounding simplified character environment, which may be an intentionally created retro art installation.

(B) The blurry handwriting on the sign looks like '湛江的', combined with the '五羊牌' ice cream, it should refer to the Zhanjiang area in western Guangdong.

(C) The sign says '广西的', this direct geographical identifier limits the scope to the Guangxi Zhuang Autonomous Region, which is a decisive textual clue.

(D) The blue small sign in the middle shop says pepsi, which is a global brand and does not have geographical orientation, thus not being the key to positioning.

**Ground Truth: (C)**

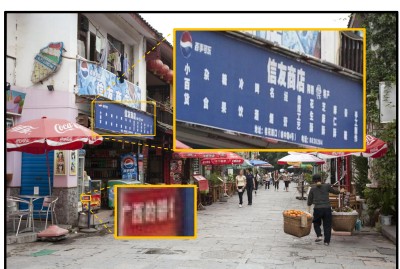

<image 1>

**Answer:** The key sign is the small board with white handwriting. Its text is blurry and cannot be read with certainty. Option A treats the reading cautiously: even if it reads as '廣西的', the use of the traditional '廣' in a mostly simplified-character environment suggests it might be a stylized/retro sign and thus should not be taken as a decisive locator. Options B and C both assert specific readings ('湛江的' or '广西的') and make strong geographic claims from unclear text, while D misidentifies a different (Pepsi) sign and ignores the prompt. Therefore, A is the most cautious inference.

**Error Reason:** The model exhibits poor fine-grained recognition capabilities and struggles to identify the most valuable subtle clues within images as effectively as humans do.

Figure 18: A sample error case of Perceptual Limitation.

# E  PROMPT TEMPLATE

---

## Question Generation Prompt Stage 1

You are a geolocation detective tasked with thoroughly analyzing all visible objects in the image and using these clues to deduce the location where the photo was taken.

**INPUT FORMAT:**
1. Problem: The original geolocation task
2. The High-resolution image

**OUTPUT FORMAT:**
JSON FORMAT:
```
[
    {
        "reasoning": "<Your overall reasoning logic for the image
          placement, along with a brief explanation of why these clues
          were selected for the question>",
        "identified_clues(Group clues of the same type
          together and indicate their locations for manual verification)":
          [
            "<Identified textual clue>: xxx,xxx,xxx",
            "<Identified second potential clue>",
            "<Identified third potential clue>",
            " ..."
          ]
    }
]
```

---

## Question Generation Prompt Stage 2

I will provide you with an image, along with its reasoning process and clues. You need to design questions for each clue (if a clue contains extensive information, you may break it down into further questions; you may also discover additional clues).

**SPECIAL NOTES:**
1. Construct options with phrasing similar to the correct answer.
2. Questions should ask things like "What is xxx like? How do the details help narrow the geographic range?" (not limited to this format). Options should use similar phrasing.
3. Questions should focus on how clues help infer location.
4. Maintain consistent length across options (avoid excessive length), otherwise test-takers may tend to choose the longest option among three shorter ones.
5. If textual clues are present, ensure every character on the sign appears in the options. For textual clues, construct similar words in options (e.g., "Brrier" and "Bmior") to maximize interference.
6. Option content must relate to the image, even for incorrect distractors.
7. Do not describe clue characteristics in the question itself. These features should reside within the options, where distractors are strategically placed to create confusion and challenge the solver!

**INPUT FORMAT:**
1. Reasoning process and clues
   JSON format:
   [
     {
       "reasoning": {reasoning},
       "identified_clues(clues of the same type grouped together, indicating their locations for manual
         verification)":
       {clues_text}
     }
   ]
2. The High-resolution image

**OUTPUT FORMAT:**
Output returns in JSON format:
[{{
    "questions":
    [{{
        "question": ... ,
        "options": {{...}},
        "answer": ... ,
        "design_rationale": ...
    }... }]
}}]

Output strictly according to the above JSON format without adding any additional text or explanations

1890
1891
1892
1893
1894
1895
1896
1897
1898
1899
1900
1901
1902
1903
1904
1905
1906
1907
1908
1909
1910
1911
1912
1913
1914
1915
1916
1917
1918
1919
1920
1921
1922
1923
1924
1925
1926
1927
1928
1929
1930
1931
1932
1933
1934
1935
1936
1937
1938
1939
1940
1941
1942
1943

**Evaluation Prompt**

[Image] [Question] The choices are listed below:
(A) [Choice A]
(B) [Choice B]
(C) [Choice C]
(D) [Choice D]
...
Analyze the image and question, then provide your reasoning process and conclusion. If there is no correct answer, explain why.
Output in JSON format:
```
[
  {
     "answer": "your analysis and your solution"
  }
]
```

**Evaluation Prompt**

**ROLE AND TASK**
You are a meticulous geolocation analysis expert. Your task now is to:
Based on the correct answers to clue questions, answer the location
question.

[Image]

**CORRECT ANSWERS TO CLUE QUESTIONS**
The following are the CORRECT answers to clue questions in this
image:
{gt_clue_summary}

**ANSWER THE LOCATION QUESTIONS**
Now, please strictly base your answers on the CORRECT judgments
regarding the clue questions listed above, and answer the location
question in this image. Select the final conclusion that best aligns
with the evidence chain from the following options.
{choices_text}

Analyze the image and question, then provide your reasoning process
and conclusion based on the correct clue evidences provided. If there
is no correct answer, explain why.

**OUTPUT FORMAT:**
Output in JSON format:
```
[
  {
    "answer": "your analysis and solution"
  }
]
```

