# OpenReview forum: "ATOM-Bench: From Atoms to Conclusions in Objective Evaluation of Large Multimodal Models Reasoning"
_ICLR.cc/2026/Conference — ICLR 2026 Conference Withdrawn Submission_

### Official Review · Reviewer_WwKZ · 2025-10-25

**Soundness:** 2
**Presentation:** 3
**Contribution:** 2
**Rating:** 2
**Confidence:** 4

**Summary:**

This paper is about the evaluation of large multimodal model reasoning. The authors claim that they introduce a CoT evaluation framework built on objective atomic questions, covering 570 high-resolution real-world images and 2,920 questions across 4 cognitive dimensions, and 12 domains, including architecture, text, transportation, culture, climate, and geology. They have tested a number of large multimodal models with the proposed evaluation framework.

**Strengths:**

1. The authors have evaluated a number of multimodal large language models with the proposed evaluation framework.
2. The proposed benchmark covers a wide range of domains.

**Weaknesses:**

1. The authors claim that "Current CoT evaluation paradigms rely on powerful LLMs as judges of free-form text, but this introduces bias and hallucination from the evaluator itself." However, the proposed evaluation framework also relies on LLMs and shares the weakness of prior works.
2. The proposed evaluation framework relies on "rigorous human review and refinement", as the authors claim. However, it is not clear how the authors ensure rigor in this process. Are there any human errors in this process? How to ensure the human experts have checked the dataset with care?
3. The evaluation framework relies heavily on human efforts in checking many details in the evaluation, which is not practical and hard to scale with paid annotators. Additionally, the proposed benchmark is relatively small. Although the authors claim it covers a wide range of fields and cognitive dimensions, I wonder whether these fields or dimensions are properly represented with limited data.
4. Given the fact that the dataset is small and the method is not practical, I doubt whether this paper has made enough contribution.

**Questions:**

Please refer to the weakness section.

---

### Official Review · Reviewer_db8N · 2025-10-28

**Soundness:** 2
**Presentation:** 3
**Contribution:** 2
**Rating:** 4
**Confidence:** 3

**Summary:**

This paper notes that while Chain-of-Thought (CoT) reasoning improves Large Multimodal Models (LMMs) in complex image-text tasks, current CoT evaluation—relying on LLMs as judges—suffers from bias, hallucination, and mispenalizing stylistic variations over real reasoning failures.
To address this, it proposes ATOM-Bench: a framework that decomposes complex tasks into atomic questions (covering 570 high-res images, 2,920 questions across 4 cognitive dimensions and 12 domains) and introduces three quantitative metrics (RCS, HI, RRS) to turn subjective evaluation into evidence-based diagnostics, solving the "black-box evaluating a black-box" issue.
Experiments on 22 LMMs show even state-of-the-art models mismatch final answer correctness with evidence comprehension and have cognitive rigidity. The paper contributes an objective, reproducible CoT evaluation framework, the first high-res process-oriented CoT benchmark, and insights into LMMs’ gaps in reasoning faithfulness and flexibility to advance reliable LMM research.

**Strengths:**

1. Instead of relying on Large Language Models (LLMs) as judges in traditional paradigms, ATOM-Bench adopts "atomic questions" to eliminate the "black-box evaluating black-box" dilemma.
2. The benchmark is built on 570 high-resolution real-world images, validated through human-machine collaboration (including expert cross-reviews of clue authenticity and distractor rationality) to guarantee data quality.
3. It decomposes complex reasoning tasks into clue-level (CLQ) and conclusion-level (CoLQ) atomic nodes, covering 4 cognitive dimensions and 12 real-world domains. T

**Weaknesses:**

1. The benchmark only centers on single-image geolocation, failing to cover complex scenarios like video temporal reasoning or cross-modal generation. It cannot fully measure LMMs’ performance across diverse CoT tasks.
2. All questions are multiple-choice, with no assessment of free-text reasoning chain generation. It cannot evaluate models’ ability to express logical steps in open text for real-world applications.
3. Complex reasoning is decomposed into pre-defined "standard chains," ignoring the diverse reasoning paths models may actually take. It fails to reflect models’ real logical decision-making processes.

**Questions:**

1. Its atomic decomposition relies on preset logic. Is this decomposition consistent with humans’ actual reasoning paths?
2. ATOM-Bench lacks samples from low-resource regions (e.g., small countries). Does it plan to supplement such data to improve evaluation comprehensiveness?
3. It doesn’t specify the weight of image vs. text clues. When clues conflict, can current metrics fairly measure models’ decision rationality?

---

### Official Review · Reviewer_sR7U · 2025-10-30

**Soundness:** 2
**Presentation:** 3
**Contribution:** 3
**Rating:** 4
**Confidence:** 4

**Summary:**

- The author proposes a novel atomic-question-based CoT evaluation framework, comparing to the previous benchmark which reasons over the CoT process using traditional "LLM as a Judge", the evaluation framework focuses on objective and fairness, including three new evaluation metrics, RCS(reasoning-conclusion support), HI(Hallucinated Inference), RRS(Reasoning Revision Score).
- Besides the evaluation framework, the authors introduce ATOM-Bench, the benchmark includes 2,920 multi-choice questions across 4 cognitive dimensions and 12 subtasks.
- Authors also evaluates 22 leading models and provide insights including even the state-of-the-art models like Gemini-2.5-Pro and GPT-5 can show post-hoc fallacies, models often fail to revise errors when confronted with indisputable ground-truth evidence.

**Strengths:**

- The originality of the paper is good, the paper focuses on the fair and objective evaluation without llm-as-the-judge process.
- The dimension of the ATOM-Bench is good, it includes 14 different atomic skills, including spatial reasoning.
- The evaluation models are sufficient, including 22 leading models with both open-sourced and close-sourced ones.
- The data curation process is very clear to the readers:
Each step clearly specifies which data sources were used, what criteria were applied, and how the results were verified.
Human verification and inter-annotator agreement evaluation were introduced to ensure annotation quality.
The logic behind the question categorization is clear and task-oriented.
- The structure of the paper is easy to read.

**Weaknesses:**

- Overall, I appreciate the readers for the presentation of this paper, however, **examples** are significantly insufficiant for both methodology part and evaluation part. And I think this is one of the biggest weakness of this paper. I search very carefully for more detailed examples in the appendix and only find failure analysis and a few failure examples.

     More specifically, authors should provide more examples regarding:
1. full multimodal reasoning process of a model regarding the answer, how to evaluate based on that example
2. examples of how atomic tasks compose into complex reasoning
3. lacks visual illustrations of multimodal input and error analysis

- The CoT evaluation framework and benchmark samples are only applied on geolocation, which according to my knowledge, this pipeline and benchmark could also be used to a broader domain for evaluating reasoning process, e.g. Mathematical Reasoning, Science Reasoning, which also needs step-by-step objective reasoning process in order to successfully answer a question.

- About the evaluation metric, the RCS (Reasoning Consistency Score) and HI (Hallucination Index) is overlapping with each other, e.g. high reasoning consistency scores indicates low hullucination index. The evaluation dimension is not diverse enough. Have you considered other evaluation metrics, for example, evaluate perception error and reasoning error separately.

- Although the benchmark fucos on objective evaluation, the structure of reasoning, the completeness and the soundness of the answer, however, are the keys to evaluate the correctness of the answer, but the paper entirely ignores them.

**Questions:**

I provide the following questions for authors:
- The motivation of the paper is to reduce the biased evaluation of "llm-as-the-judge", however, in the evaluation metrics, there is some human threshold. e.g. In RCS, the τ=0.75 is set by human. This step also introduces human bias. Why is τ=0.75? Do you have explanations on it?
- The author claims that the proposed evaluation metric is more objective and fair than traditional evaluation method. Do you have quantitative results to prove that? For example, sample a subset of Atom-Bench and using standard "llm-as-the-judge" process to evaluate and compare it against the proposed evaluation metrics.
- Can the proposed evaluation framework generalize to other domains except for the geographical reasoning task, e.g. for a broader domain, e.g. in Mathematical Reasoning? If so, could you provide examples regarding how to apply this framework into a broder domain, e.g. Mathematical Reasoning using a standard Benchmark like Mathvista?
- **Especially**, you should provide more examples regarding: (as listed in weakness)
1. full multimodal reasoning process of a model regarding the answer, how to evaluate based on that example
2. examples of how atomic tasks compose into complex reasoning
3. visual illustrations of multimodal input and error analysis

I will consider raise my score if you can address my concerns listed above.

---

### Official Review · Reviewer_sAqG · 2025-11-01

**Soundness:** 2
**Presentation:** 3
**Contribution:** 2
**Rating:** 2
**Confidence:** 4

**Summary:**

The paper presents ATOM-Bench, a benchmark for evaluating reasoning processes in large multimodal models. It reformulates complex reasoning into atomic multiple-choice questions with ground-truth answers to enable objective measurement. The dataset contains 570 real-world images and 2,920 questions covering four cognitive dimensions and twelve subdomains. The authors introduce three metrics to assess reasoning consistency, hallucination rate, and robustness when models are given corrected evidence. Experiments on 22 multimodal models are conducted to analyze their reasoning behavior and the relationship between answer accuracy and reasoning consistency.

**Strengths:**

1. Atomic multiple-choice questions provides objective, interpretable, and reproducible evaluation results.
2. The analysis highlights a clear gap between correctness and reasoning quality, offering concrete empirical observations.

**Weaknesses:**

1. The dataset is relatively small and limited in diversity, containing only 570 real-world images, which restricts coverage of varied visual scenes and reduces the stability of cross-model comparisons.
2. The task scope is overly narrow, as the framework is primarily validated on single-image geo-localization, limiting its generalizability to other multimodal reasoning tasks.
3. Error analysis remains anecdotal and lacks systematic statistics on error types or cross-model differences, making it difficult to derive actionable insights for model improvement.

**Questions:**

1. Could it be extended to cover more complex or diverse multimodal reasoning tasks beyond single-image geolocation?
2. Could the authors provide a more detailed error analysis with statistics and a deeper look at failure patterns?

---

### Note · Authors · 2025-11-15

I have read and agree with the venue's withdrawal policy on behalf of myself and my co-authors.